



# Evaluation of bacterial glycerol dialkyl glycerol tetraether and $^2$H-$^{18}$O biomarker proxies along a Central European topsoil transect

Johannes Hepp[1,2,*], Imke K. Schäfer[3], Verena Lanny[4], Jörg Franke[3], Marcel Bliedtner[3,a], Kazimierz Rozanski[5], Bruno Glaser[2], Michael Zech[2,6], Timothy I. Eglinton[4], Roland Zech[3,a]

[1]Chair of Geomorphology and BayCEER, University of Bayreuth, 95440 Bayreuth, Germany and

[2]Institute of Agronomy and Nutritional Sciences, Soil Biogeochemistry, Martin-Luther-University Halle-Wittenberg, 06120 Halle, Germany

[3]Institute of Geography and Oeschger Centre for Climate Change Research, University of Bern, 3012 Bern, Switzerland

[4]Department of Earth Science, ETH Zurich, 8092 Zurich, Switzerland

[5]Faculty of Physics and Applied Computer Science, AGH University of Science and Technology, 30-059 Kraków, Poland

[6]Institute of Geography, Faculty of Environmental Sciences, Technical University of Dresden, 01062 Dresden, Germany

[a]now at Institute of Geography, Chair of Physical Geography, Friedrich-Schiller University of Jena, 07743 Jena, Germany

*corresponding author (johannes-hepp@gmx.de)



## Keywords

Leaf wax $n$-alkanes, hemicellulose sugars, pH, temperature, CBT, MBT', precipitation $\delta^2H/\delta^{18}O$, relative humidity

## Abstract

Molecular fossils, like bacterial branched glycerol dialkyl glycerol tetraethers (brGDGTs), and the stable isotopic composition of biomarkers, such as $\delta^2H$ of leaf wax-derived $n$-alkanes ($\delta^2H_{n\text{-}alkane}$) or $\delta^{18}O$ of hemicellulose-derived sugars ($\delta^{18}O_{sugar}$) are increasingly used for the reconstruction of past climate and environmental conditions. Plant-derived $\delta^2H_{n\text{-}alkane}$ and $\delta^{18}O_{sugar}$ values record the isotopic composition of plant source water ($\delta^2H/\delta^{18}O_{source\text{-}water}$), which usually reflects mean annual precipitation ($\delta^2H/\delta^{18}O_{precipiation}$), modulated by evapotranspirative leaf water enrichment and biosynthetic fractionation. Accuracy and precision of respective proxies should be ideally evaluated at a regional scale. For this study, we analysed topsoils below coniferous and deciduous forests, as well as grassland soils along a Central European transect in order to investigate the variability and robustness of various proxies, and to identify effects related to vegetation. Soil pH-values derived from brGDGTs correlate reasonably well with measured soil pH-values, but systematically overestimate them ($\Delta pH = 0.6 \pm 0.6$). The branched vs. isoprenoid tetraether index (BIT) can give some indication whether the pH reconstruction is reliable. Temperatures derived from brGDGTs overestimate mean annual air temperatures slightly ($\Delta T_{MA} = 0.5°C \pm 2.4$). Apparent isotopic fractionation ($\varepsilon_{n\text{-}alkane/precipitation}$ and $\varepsilon_{sugar/precipitation}$) is lower for grassland sites than for forest sites due to "signal damping", i.e. grass biomarkers do not record the full evapotranspirative leaf water enrichment. Coupling $\delta^2H_{n\text{-}alkane}$ with $\delta^{18}O_{sugar}$ allows to reconstruct the stable isotopic composition of the source water more accurately than without the coupled approach ($\Delta\delta^2H = \sim\!-21‰ \pm 22$ and $\Delta\delta^{18}O = \sim\!-2.9‰ \pm 2.8$). Similarly, relative humidity during daytime and vegetation period (RH_{MDV}) can be reconstructed using the coupled isotope approach ($\Delta RH_{MDV} = \sim\!-17 \pm 12$). Especially for coniferous sites, reconstructed RH_{MDV} values as well as source water isotope composition underestimate the measured values. This can be likely explained by understory grass vegetation at the coniferous sites contributing significantly to the $n$-alkane pool but only marginally to the sugar pool in the topsoil. The large uncertainty likely reflect the fact that biosynthetic fractionation is not constant, as well as microclimate variability. Overall, GDGTs and the coupled $\delta^2H_{n\text{-}alkane}$-$\delta^{18}O_{sugar}$ approach have great potential for more quantitative paleoclimate reconstructions.





## 1 Introduction

Information about the variability and consequences of past climate changes is a prerequisite for precise predictions regarding the present climate change. Molecular fossils, so called biomarkers, climate proxies have great potential to enhance our understanding about variations of past climate and environmental changes. Lipid biomarkers in particular, are increasingly used for paleoclimate and environmental reconstructions (e.g. Brincat et al., 2000; Eglinton and Eglinton, 2008; Rach et al., 2014; Romero-Viana et al., 2012; Schreuder et al., 2016). However strengths and limitations of respective proxies need known (Dang et al., 2016). For this, calibrations using modern reference samples are essential.

Terrestrial branched glycerol dialkyl glycerol tetraethers (brGDGTs) that are synthesized in the cell membranes of anaerobe heterotrophic soil bacteria (Oppermann et al., 2010; Weijers et al., 2010) have great potential for the reconstruction of past environmental conditions (e.g. Coffinet et al., 2017; Schreuder et al., 2016; Zech et al., 2012), although some uncertainties exist. Calibration studies suggest that the relative abundance of the individual brGDGTs varies with mean annual air temperature ($T_{MA}$) and soil pH (Peterse et al., 2012; Weijers et al., 2007), at least across large, global climate gradients or along pronounced altitudinal gradients (Wang et al., 2017). However, in arid regions the production of brGDGT is limited, while isoprenoidal GDGTs (iGDGTs) produced by archaea provide the dominant part of the overall soil GDGT pool (Anderson et al., 2014; Dang et al., 2016; Dirghangi et al., 2013; Wang et al., 2013; Xie et al., 2012). The ratio of brGDGTs vs. isoprenoid GDGTs (BIT) can be used as indication whether a reconstruction of $T_{MA}$ and pH will be reliable. Moreover, Mueller-Niggemann et al. (2016) revealed an influence of the vegetation cover on the brGDGT producing soil microbes. From field experiments, it is known, that vegetation type and mulching practice strongly effect soil temperature and moisture (Awe et al., 2015; Liu et al., 2014). Thus, multiple factors can be expected to influence soil microbial communities and GDGT production. So far, little is known about the variability of GDGT proxies on a regional scale, and a calibration study with small climate gradient but with different vegetation types might be useful.

Compound specific stable hydrogen isotopes of leaf wax biomarkers, such as long chain $n$-alkanes ($\delta^2H_{n\text{-alkanes}}$) record the isotopic signal of precipitation and therefore past climate and environmental conditions (Sachse et al., 2004, 2006). However, various influencing factors are known all along the way from the moisture source to leaf waxes (Pedentchouk and Zhou, 2018 and Sachse et al., 2012 for review). One is the evapotranspiration of leaf water (Feakins and Sessions, 2010; Kahmen et al., 2013; Zech et al., 2015), which is strongly driven by relative air humidity (RH; e.g. Cernusak et al., 2016 for review). In addition, a strong precipitation signal is known to be incorporated into long chain leaf waxes (Hou et al., 2008; Rao et al., 2009; Sachse et al., 2004). In paleoclimate studies, it is often not feasible to disentangle between the evapotranspirative enrichment from the precipitation signal. Zech et al. (2013) proposed to couple $\delta^2H_{n\text{-alkane}}$ results with oxygen stable isotopes of hemicellulose-derived sugars ($\delta^{18}O_{sugar}$). Assuming constant biosynthetic fractionation factors ($\varepsilon_{bio}$) for the different compound classes ($n$-alkanes and hemicellulose sugars), the coupling enables the reconstruction of the isotopic composition of leaf water, RH and $\delta^2H/\delta^{18}O$ of plant source water ($\approx \delta^2H/\delta^{18}O$ of precipitation; Tuthorn et al., 2015). So far, a detailed evaluation of this approach on the European scale, as well as concerning possible effects related to vegetation changes is missing.



We analysed topsoil samples under coniferous, deciduous and grassland vegetation along a Central European transect in order to estimate the variability of the biomarker proxies. More specifically, we aim to test whether:

(i) the vegetation type has an influence on the brGDGT proxies, the $\delta^2H_{n\text{-alkane}}$ and the $\delta^{18}O_{sugar}$ stable isotopic composition, as well as on reconstructed $\delta^2H/\delta^{18}O_{source\text{-water}}$ and RH.

(ii) the published brGDGT proxies used for reconstructing mean annual temperature and soil pH are sensitive enough to reflect the medium changes in temperature and soil pH along our transect.

(iii) the coupled $\delta^2H_{n\text{-alkane}}$-$\delta^{18}O_{sugar}$ approach faithfully reflects $\delta^2H/\delta^{18}O$ of precipitation and RH along the transect.

## 2 Material and methods

### 2.1 Geographical setting and sampling

In November 2012, we collected topsoil samples (0-5 cm depth) at 16 locations along a transect from Southern Germany to Southern Sweden (Fig. 1A) and distinguished between sites with coniferous forest (con, n = 9), deciduous forest (dec, n = 14) and grassland (grass, n = 6) vegetation cover (for more details see Schäfer et al. (2016) and Tab. S1).

### 2.2 Database of instrumental climate variables and isotope composition of precipitation

Climate data was derived from close-by weather observation stations operating by the regional institutions (Deutscher Wetterdienst (DWD) for Germany, Danmarks Meteorologiske Institut (DMI) for Denmark and the Sveriges Meteorologiska och Hydrologiska Institute (SMHI) for Sweden). The DWD provides hourly data for each station (DWD Climate Data Center, 2018b), enabling not only the calculation of $T_{MA}$, but also of the mean annual relative air humidity ($RH_{MA}$), mean temperature and relative air humidity during the vegetation period ($T/RH_{MV}$), and of daytime temperature and relative humidity averages over the vegetation period ($T/RH_{MDV}$). In addition, annual precipitation observations were used to derive the mean annual precipitation amount ($P_{MA}$; DWD Climate Data Center, 2018b). From the DMI, the respective climate variables were derived from published technical reports (Cappelen, 2002; Frich et al., 1997; Laursen et al., 1999). The SMHI provides open data from which we derived the climate variables for the Swedish sites (Swedish Meteorological and Hydrological Institute, 2018). For more details about the climate database used for calculations and comparisons, the reader is referred to Tab. S2.

For comprising German precipitation $\delta^2H/\delta^{18}O$ along the transect, we realized a regionalisation (called $\delta^2H/\delta^{18}O_{GIPR}$) using online available data from 34 German GNIP stations, 4 Austrian ANIP stations and the Groningen GNIP station (van Geldern et al., 2014; IAEA/WMO, 2018; Stumpp et al., 2014; Umweltbundesamt GmbH, 2018), following the approach of Schlotter (2007). However, instead of the multivariate regression procedure applied by Schlotter (2007), we used a random forest approach (Hothorn et al., 2006; Strobl et al., 2007, 2008) to describe the relationship of squared latitude, latitude, longitude and altitude vs. long term weighted means of precipitation $\delta^2H/\delta^{18}O$, and realized the prediction for the study sites. For the Danish





and Swedish sites, such a procedure was not possible. Hence, the annual precipitation $\delta^2H/\delta^{18}O$
values were derived from the Online Isotopes in Precipitation Calculator (OIPC, version 3.1),
therefore called $\delta^2H/\delta^{18}O_{OIPC}$ (Bowen, 2018; Bowen and Revenaugh, 2003; IAEA/WMO,
2015). The finally used $\delta^2H/\delta^{18}O_{GIPR/OIPC}$ data are given in Tab. S1.
The $T_{MA}$ along the transect ranges from 5.3 to 10.6°C, and $P_{MA}$ ranges from 554 to 1769 mm
(Fig. 1B). Precipitation $\delta^2H/\delta^{18}O$ shows moderate changes along the transect, $\delta^2H_{GIPR/OIPC}$
varies between -52 and -79‰, and $\delta^{18}O_{GIPR/OIPC}$ ranges from -7.4 to -10.9‰ (Fig. 1C).
Correlations between $\delta^{18}O_{GIPR/OIPC}$ and $P_{MA}$, altitude of the locations, $T_{MA}$ are given in the
supplementary material (Fig. S1 to S3), along with a $\delta^2H_{GIPR/OIPC}$ vs. $\delta^{18}O_{GIPR/OIPC}$ scatter plot
(Fig. S4).

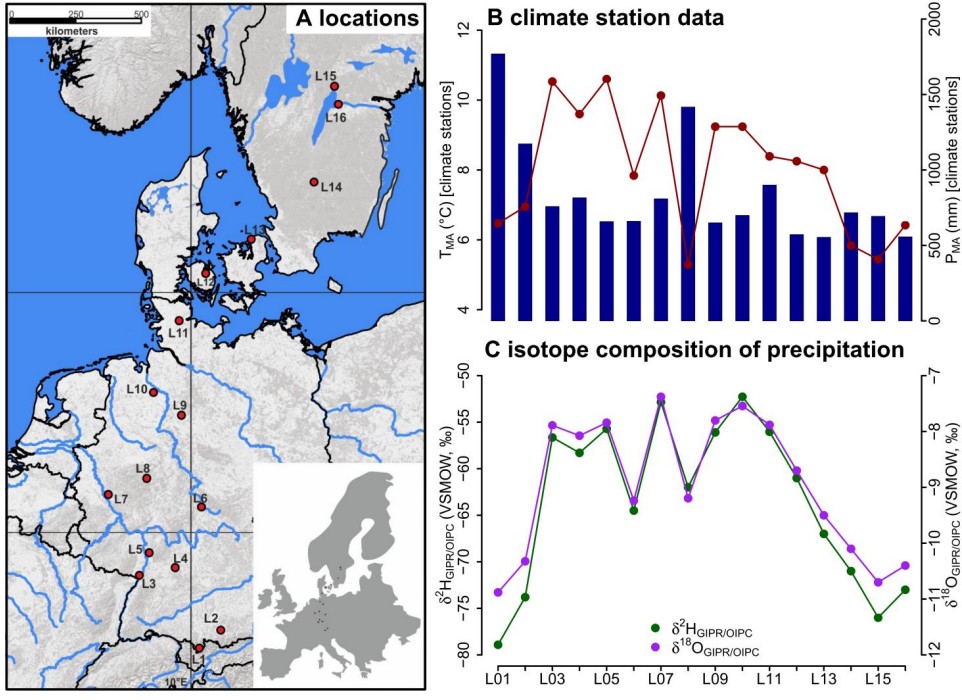


**Fig. 1.** (A) Sample locations (red dots, map source: US National Park Service), (B) variations
of mean annual air temperature ($T_{MA}$) and mean annual precipitation ($P_{MA}$) derived from close-
by climate station data, and (C) hydrogen and oxygen stable isotope composition of
precipitation ($\delta^2H_{GIPR/OIPC}$ and $\delta^{18}O_{GIPR/OIPC}$, respectively) as derived for the sampled transect
locations (see section 2.2 GIPR $\delta^2H/\delta^{18}O$ generation procedure). The reader is referred to
section 2.2 (and Tab. S1 and S2) for database and reference information of data plotted in (B)
and (C).

**2.3 Soil extractions and analysis**
2.3.1 GDGTs and pH
A detailed description of sample preparation for lipid analysis can be found in Schäfer et al.
(2016). Briefly, 1–6 g freeze-dried and grounded soil sample was microwave extracted with 15





ml dichloromethane (DCM)/methanol (MeOH) 9:1 *(v:v)* at 100°C for 1 h. Extracts were
separated over aminopropyl silica gel (Supelco, 45 μm) pipette columns. The nonpolar fraction
(including *n*-alkanes) was eluted with hexane and further purified over $AgNO_3$ coated silica
pipette columns (Supelco, 60-200 mesh) and zeolite (Geokleen Ltd.). The GDGT-containing
fraction was eluted with DCM:MeOH 1:1 *(v:v)*, re-dissolved in hexane/isopropanol (IPA) 99:1
*(v:v)* and transferred over 0.45 μm PTFE filters into 300 μl inserts. For quantification, a known
amount of a $C_{46}$ diol standard was added after transfer. The samples were analysed at ETH
Zurich using an Agilent 1260 Infinity series HPLC–atmospheric chemical pressure ionization
mass spectrometer (HPLC–APCI-MS) equipped with a Grace Prevail Cyano column (150 mm
× 2.1 mm; 3 μm). The GDGTs were eluted isocratically with 90% A and 10% B for 5 min and
then with a linear gradient to 18% B for 34 min at 0.2 ml min$^{-1}$, where A=hexane and
B=hexane/isopropanol (9:1, *v:v*). Injection volume was 10 μl and single ion monitoring of
$[M+H]^+$ was used to detect GDGTs.
The pH of the samples was measured in the laboratory of the Soil Biogeochemistry group,
Institute of Agronomy and Nutritional Sciences, Martin-Luther-University Halle-Wittenberg,
in a 1:3 soil:water *(w/v)* mixture.

2.3.2 $\delta^2H_{n\text{-alkane}}$
The hydrogen isotopic composition of the highest concentrated *n*-alkanes (*n*-$C_{25}$, *n*-$C_{27}$, *n*-$C_{29}$,
*n*-$C_{31}$, and *n*-$C_{33}$) was determined using a TRACE GC Ultra Gas Chromatography connected to
a Delta V Plus Isotope Ratio Mass Spectrometer via a $^2H$ pyrolysis reactor (GC-$^2H$-Py-IRMS;
Thermo Scientific, Bremen, Germany) at the ETH Zurich. The compound-specific $^2H/^1H$ ratios
were calibrated against an external standard with $C_{15} - C_{35}$ homologues. External standard
mixtures (A4 mix from A. Schimmelmann, University of Indiana) were run between the
samples for multipoint linear normalization. The $H^+_3$ factor was determined on each
measurement day and was constant throughout the periods of the sample batches. Samples were
analysed in duplicates, and results typically agreed within 4‰ (average difference = 1.4‰). All
$\delta^2H$ values are expressed relative to the Vienna Standard Mean Ocean Water (V-SMOW).

2.3.3 $\delta^{18}O_{sugar}$
Hemicellulose sugars were extracted and purified using a slightly modified standard procedure
(Amelung et al., 1996; Guggenberger et al., 1994; Zech and Glaser, 2009). Briefly, myoinositol
was added to the samples prior to extraction as first internal standard. The sugars were released
hydrolytically using 4M trifluoroacetic acid for 4 h at 105°C, cleaned over glass fiber filters and
further purified using XAD and Dowex columns. Before derivatization with methylboronic acid
(Knapp, 1979), the samples were frozen, freeze-dried, and 3-O-methylglucose in dry pyridine
was added as second internal standard. Compound-specific hemicellulose sugar $^{18}O$
measurements were performed in the laboratory of the Soil Biogeochemistry group, Institute of
Agronomy and Nutritional Sciences, Martin-Luther-University Halle-Wittenberg, using GC-
$^{18}O$-Py-IRMS (all devices from Thermo Fisher Scientific, Bremen, Germany). Standard
deviations of the triplicate measurements were 1.4‰ (over 29 investigated samples) for
arabinose and xylose, respectively. We focus on these two hemicellulose-derived neutral sugars





arabinose and xylose as they strongly predominate over fucose in terrestrial plants, soils and
sediments (Hepp et al., 2016 and references therein). Rhamnose concentrations were too low to
obtain reliable $\delta^{18}O$ results. All $\delta^{18}O$ values are expressed relative to the Vienna Standard Mean
Ocean Water (V-SMOW).

**2.4 Theory and Calculations**
2.4.1 Calculations used for the GDGT-based reconstructions
The branched and isoprenoid tetraether (BIT) index is calculated according to Hopmans et al.
(2004), for structures see Fig. S5:
$$BIT = \frac{Ia+IIa+IIIa}{Ia+IIa+IIIa+crenarchaeol}. \tag{1}$$
The cyclopentane moiety number of brGDGTs correlates negatively with soil pH (Weijers et
al., 2007), which led to the development of the cyclization of branched tetraethers (CBT) ratio.
CBT and the CBT based pH (pH$_{CBT}$) were calculated according to Peterse et al. (2012):
$$CBT = -\log \frac{Ib+IIb}{Ia+IIa}, \tag{2}$$
$$pH_{CBT} = 7.9 - 1.97 \times CBT. \tag{3}$$
The number of methyl groups in brGDGTs correlates negatively with T$_{MA}$ and soil pH (Peterse
et al., 2012; Weijers et al., 2007). Thus, the ratio of the methylation of branched tetraethers
(MBT) ratio and the CBT ratio can be used to reconstruct T$_{MA}$. We use the equation given by
Peterse et al. (2012):
$$MBT' = \frac{Ia+Ib+Ic}{Ia+Ib+Ic+IIa+IIb+IIc+IIIa}, \tag{4}$$
$$T_{MA} = 0.81 - 5.67 \times CBT + 31.0 \times MBT'. \tag{5}$$

2.4.2 Calculations and concepts used for the coupled $\delta^2H$-$\delta^{18}O$ approach
The apparent fractionation is calculated according to Cernusak et al. (2016):
$$\varepsilon_{n\text{-alkane/precipitation}} = \left( \frac{\delta^2H_{n\text{-alkane}} - \delta^2H_{GIPR/OIPC}}{1 + \delta^2H_{GIPR/OIPC}/1000} \right), \tag{6}$$
$$\varepsilon_{sugar/precipitation} = \left( \frac{\delta^{18}O_{sugar} - \delta^{18}O_{GIPR/OIPC}}{1 + \delta^{18}O_{GIPR/OIPC}/1000} \right). \tag{7}$$
The isotopic composition of leaf water ($\delta^2H$/$\delta^{18}O_{leaf\ water}$) can be calculated using $\varepsilon_{bio}$ for $\delta^2H_{n\text{-}}$
$_{alkane}$ (-160‰, Sachse et al., 2012; Sessions et al., 1999) and $\delta^{18}O_{sugar}$ (+27‰, Cernusak et al.,
2003; Schmidt et al., 2001):
$$\delta^2H_{leaf\ water} = \left( \frac{1000 + \delta^2H_{n\text{-alkane}}}{1000 + \varepsilon_{bio}\ (n\text{-alkane})} \right) \times 10^3 - 1000, \tag{8}$$
$$\delta^{18}O_{leaf\ water} = \left( \frac{1000 + \delta^{18}O_{sugar}}{1000 + \varepsilon_{bio}\ (sugar)} \right) \times 10^3 - 1000. \tag{9}$$
Zech et al. (2013) introduced the conceptual model for the coupled $\delta^2H_{n\text{-alkane}}$-$\delta^{18}O_{sugar}$ approach
in detail. Briefly, the coupled approach is based on the following assumptions (illustrated in
Fig. 8): (i) The isotopic composition of precipitation, which is set to be equal to the plant source
water, typically plots along the global meteoric water line (GMWL; $\delta^2H = 8 \times \delta^{18}O + 10$) in a





$\delta^{18}O$ vs. $\delta^2H$ space (Craig, 1961); (ii) Source water uptake by plants does not lead to any
fractionation (e.g. Dawson et al., 2002), and significant evaporation of soil water can be
excluded; (iii) Evapotranspiration leads to enrichment of the remaining leaf water along the
local evaporation line (LEL; Allison et al., 1985; Bariac et al., 1994; Walker and Brunel, 1990),
compared to the source water taken up by the plant; (iv) The biosynthetic fractionation is
assumed to be constant. In addition, isotopic equilibrium between plant source water (~
weighted mean annual precipitation) and the local atmospheric water vapour is assumed.
Further assumption concerns the isotope steady-state in the evaporating leaf water reservoir.
The coupled approach allows for reconstructing the isotopic composition of plant source water
($\delta^2H/\delta^{18}O_{source\text{-}water}$) from the reconstructed leaf water, by calculating the intercepts of the LELs
with the GMWL (Zech et al., 2013). The slope of the LEL ($S_{LEL}$) can be assessed by the
following equation (Gat, 1971):
$\qquad S_{LEL} = \dfrac{\varepsilon_2^* + C_k^2}{\varepsilon_{18}^* + C_k^{18}},$             (10)
where $\varepsilon^*$ are equilibrium isotope fractionation factors and $C_k$ are kinetic fractionation factors.
The latter equals to 25.1‰ and 28.5‰, for $C_k^2$ and $C_k^{18}$, respectively (Merlivat, 1978). The
equilibrium fractionation factors can be derived from empirical equations (Horita and
Wesolowski, 1994) by using $T_{MDV}$ values. For two Danish sites $T_{MDV}$ are not available, instead
$T_{MV}$ is used here (section 2.2 and Tab. S2).
In a $\delta^{18}O$-$\delta^2H$ diagram, the distance of the leaf water from the GMWL define the deuterium-
excess of leaf water ($d_{leaf\text{-}water} = \delta^2H_{leaf\text{-}water} - 8 \times \delta^{18}O_{leaf\text{-}water}$, according Dansgaard, (1964); Fig.
8). To convert $d_{leaf\text{-}water}$ into mean RH during daytime and vegetation period ($RH_{MDV}$), a
simplified Craig-Gordon model can be applied (Zech et al., 2013):
$RH = 1 - \dfrac{\Delta d}{\varepsilon_2^* - 8 \times \varepsilon_{18}^* + C_k^2 - 8 \times C_k^{18}},$          (11)
where $\Delta d$ is the difference in $d_{leaf\text{-}water}$ and the deuterium-excess of source water ($d_{source\text{-}water}$).

**2.5 Statistics**
In the statistical analysis we checked sample distributions for normality (Shapiro and Wilk,
1965) and for equal variance (Levene, 1960). If normality and equal variances are given, we
perform an Analysis of Variance (ANOVA). If that is not the case, we conduct the non-
parametric Kruskal-Wallis Test. ANOVA or Kruskal-Wallis are used to find significant
differences (a=0.05) between the vegetation types (deciduous, conifer and grass).
In order to describe the relation along a 1:1 line, the coefficient of correlation ($R^2$) was
calculated as $R^2 = 1 - \sum (\text{modeled - measured})^2 / \sum(\text{measured - measured mean})^2$. The small
$r^2$ is taken as coefficient of correlation of a linear regression between a dependent (y) and
explanatory variable(s). The root mean square error (RMSE) of the relationships was calculated
as $RMSE = \sqrt{\left(\frac{1}{n} \cdot \sum(\text{modeled - measured})^2\right)}$. All data plotting and statistical analysis was
realized in R (version 3.2.2; R Core Team, 2015).



## 3 Results and Discussion

### 3.1 GDGT concentrations

GDGT Ia has the highest concentration under all vegetation types, followed by GDGT IIa and GDGT IIIa (Fig. 2). GDGT Ib, IIb and Ic occur in minor, GDGT IIc and IIIb only in trace amounts. GDGT IIIc was below the detection limit in most of the samples (Tab. S3). Although other studies document an influence of the vegetation cover on soil temperature and soil water content, which control the microbial community composition in soils (Awe et al., 2015; Liu et al., 2014; Mueller-Niggemann et al., 2016), we find no statistically different pattern of the individual brGDGTs.

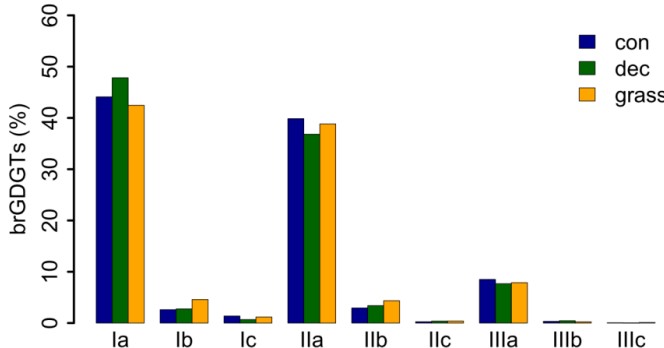

**Fig. 2.** Mean concentrations of individual brGDGTs as percentage of all brGDGTs for the three investigated types. Abbreviations: con = coniferous forest sites (n=9); dec = deciduous forest sites (n=14); grass = grassland sites (n=6).

Total concentrations of brGDGTs range from 0.32 to 9.17 µg/g dry weight and tend to be highest for the coniferous samples and lowest for the grasses (Fig. 3A, Tab. S3). Bulk brGDGT concentrations lie within ranges of other studies examining soils of mid latitude regions (Huguet et al., 2010b, 2010a; Weijers et al., 2011). Similar concentrations in coniferous and deciduous samples imply that brGDGT production does not strongly vary in soils below different forest types. The grass samples show lower brGDGT concentrations compared to the forest samples, but this is probably mainly due to ploughing of the grass sites and hence admixing of mineral subsoil material. Anyhow, the differences in brGDGT concentrations are not significant (p-value = 0.06).

### 3.2 BIT index

Most of the samples have a BIT index higher than 0.9 (Fig 3B and Tab. S3). The BIT-values are typical for soils in humid and temperate climate regions (Weijers et al., 2006). However, outliers exist. The most likely source of iGDGTs in soils are Thaumarchaeota, i.e. aerobe ammonia oxidizing archaea producing Crenarchaeol and its regioisomer (Schouten et al., 2013 and references therein), precipitation amounts drop below 700-800 mm (Dang et al., 2016; Dirghangi et al., 2013). The $P_{MA}$ data of our sampling sites mostly show precipitation > 550 mm (Fig. 1B), but one has to be aware that this data is based on the climate station nearest to the respective sampling locations and microclimate effects, such as sunlight exposure, canopy



cover or exposition might have a pronounced influence on the brGDGT vs. iGDGT distribution.
Mueller-Niggemann et al. (2016) found higher BIT indices in upland soils compared to paddy
soils and stated that the management type also influences BIT values in soils. Along our
transect, grass sites tend to have slightly lower BIT-values than forest sites, probably due to the
absence of a litter layer and hence, no isolation mechanism preventing evaporation of soil water.
Anyhow, differences between vegetation types are not significant (p-value = 0.32).

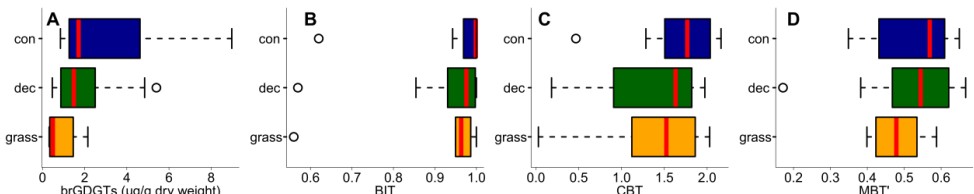


**Fig. 3.** (A) Total concentrations of brGDGTs in µg g$^{-1}$ dry weight, as well as (B) BIT, (C) CBT
and (D) MBT'. Abbreviations: con = coniferous forest sites (n=9); dec = deciduous forest sites
(n=14); grass = grassland sites (n=6). Box plots show median (red line), interquartile range
(IQR) with upper (75%) and lower (25%) quartiles, lowest whisker still within 1.5IQR of lower
quartile, and highest whisker still within 1.5IQR of upper quartile, dots mark outliers.

### 3.3 CBT-derived pH

The CBT ratio shows a pronounced variation independent of vegetation type with values
between 0.03 and 2.16 (Fig 3C). The coniferous samples tend to be highest, but the differences
between vegetation types are not significant (p-value = 0.48). The CBT index can be related to
pH in acidic and/or humid soils (e.g. Dirghangi et al., 2013; Mueller-Niggemann et al., 2016;
Peterse et al., 2012; Weijers et al., 2007) but might be an indicator of soil water content and
hence, precipitation in more arid and alkaline soils (e.g. Dang et al., 2016). There is a
pronounced correlation between CBT and soil pH (Fig. 4), which is in good agreement with
other studies from mid latitude regions where precipitation is relatively high (Anderson et al.,
2014 and references therein). Moreover, the CBT to pH relationship in terms of slope and
intersect in our dataset (CBT = -0.47 × pH + 3.5, r$^2$ = 0.7, p-value < 0.0001, n = 29) is well
comparable to the correlation described for the global calibration dataset of Peterse et al. (2012)
(CBT = -0.36 × pH + 3.1, r$^2$ = 0.7, p-value < 0.0001, n = 176).
However, there are some outliers in the CBT-pH correlation, which need a further examination
(see locations grass L04, dec L10 and dec L12 as marked in Figs. 4 and 5). The outliers show
lower BIT indices (< 0.85, Tab. S3). Even though the data from the nearest climate station
suggest no abnormal P$_{MA}$. Local effects such as differences in the amount of sunlight exposure,
nutrient availability for brGDGT producing organisms or, most likely soil water content might
influence the brGDGT production at these locations (Anderson et al., 2014; Dang et al., 2016).
A lower BIT index as well as a lower CBT occur when soil water content decreases (Dang et
al., 2016; Sun et al., 2016) or when aeration is high and less anoxic microhabitats for GDGT
producing microbes exist (e.g. Dirghangi et al., 2013).

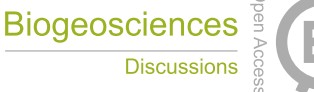

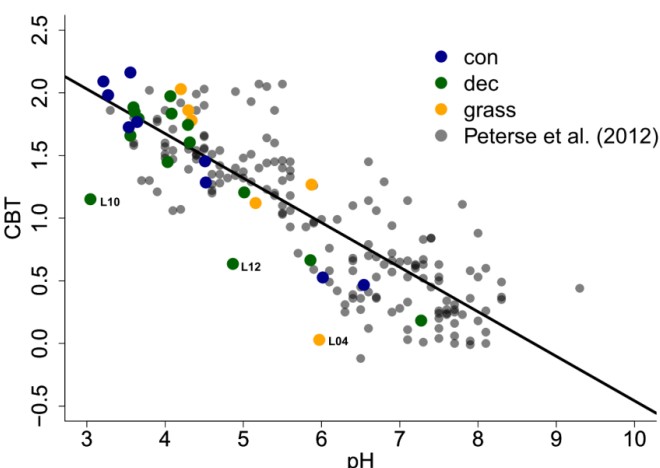


**Fig. 4.** CBT to pH relationship in our dataset in comparison to the global calibration dataset
from Peterse et al. (2012) (CBT = -0.36 × pH + 3.1, $r^2$ = 0.7, p-value < 0.0001, n = 176, black
line). Abbreviations: con = coniferous forest sites (n=9); dec = deciduous forest sites (n=14);
grass = grassland sites (n=6).

346

As the CBT and pH are similarly correlated in our dataset and the global dataset of Peterse et
al. (2012), the CBT-derived pH correlated well with the actual pH (Fig. 5A; $R^2$ = 0.3).
Expressed as ΔpH (CBT-derived pH - measured pH), there is a tendency that the GDGTs result
in an overestimation of the real pH for the forest sites (Fig. B). Yet a Kruskal-Wallis test shows
no statistically significant difference between the vegetation types, with a p-value of 0.13. The
overall ΔpH of 0.6 ±0.6 shows that the reconstruction of soil pH using brGDGTs works well
along this transect.

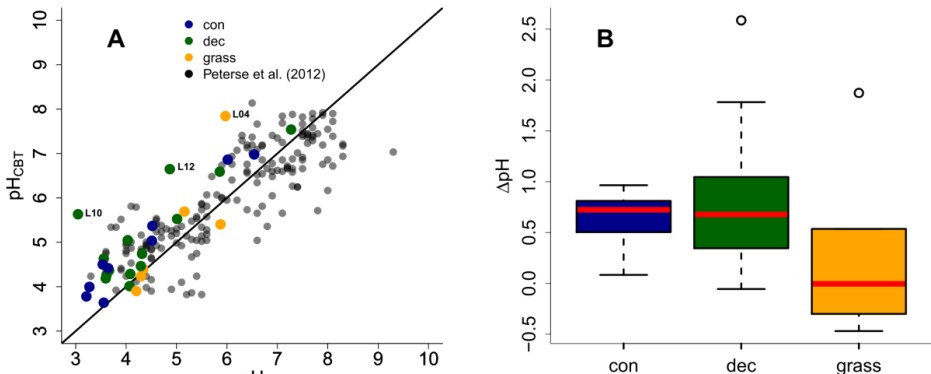

354

**Fig. 5.** (A) Correlation between measured pH and reconstructed soil pH (pH_CBT) from our
transect data in comparison to the global calibration dataset from Peterse et al. (2012) ($R^2$ = 0.7,
RMSE = 0.75, n = 176). Black line indicates the 1:1 relationship. (B) Boxplots of ΔpH (refers
to pH_CBT-pH). Box plots show median (red line), interquartile range (IQR) with upper (75%)



and lower (25%) quartiles, lowest whisker still within 1.5IQR of lower quartile, and highest
whisker still within 1.5IQR of upper quartile, dots mark outliers. Abbreviations: con =
coniferous forest sites (n=9); dec = deciduous forest sites (n=14); grass = grassland sites (n=6).

### 3.4 MBT'-CBT-derived $T_{MA}$ reconstructions

The MBT' shows high variability with values ranging from 0.17 to 0.67 no statistical
differences between vegetation types (p-value = 0.54; Fig. 3D, Tab. S3). When comparing
reconstructed (MBT'-CBT-derived) $T_{MA}$ with climate station $T_{MA}$, the data plot close to the 1:1
line, and fit well into the global dataset of Peterse et al. (2012) (Fig. 7A). The $\Delta T_{MA}$ reveal an
overall offset of 0.5°C ±2.4 and there is no statistically difference between vegetation types
(Fig. 7B). The standard deviation in $\Delta T_{MA}$ of ±2.4 is well in line with the RMSE of 5.0 for the
global calibration dataset (Peterse et al., 2012).

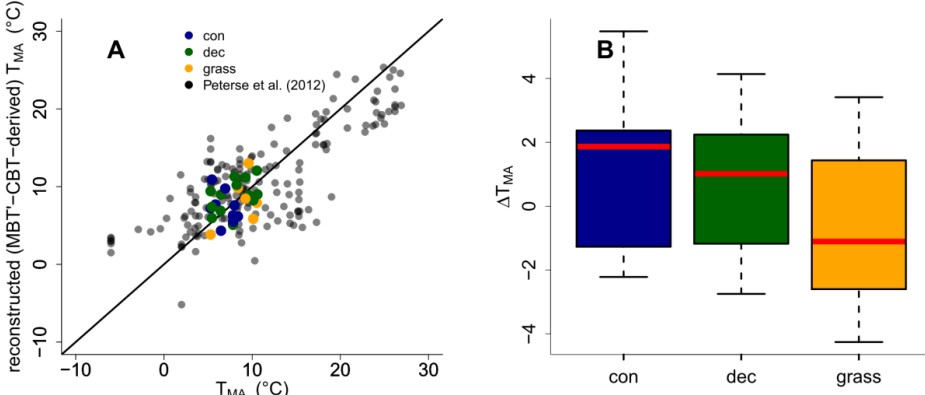

**Fig. 6.** (A) Correlation between climate station $T_{MA}$ and reconstructed (MBT'-CBT-derived)
$T_{MA}$. For comparison, the global calibration dataset from Peterse et al. (2012) is shown. The
black line indicates the 1:1 relationship. (B) Boxplots of $\Delta T_{MA}$ (refers to reconstructed $T_{MA}$-
$T_{MA}$ from climate stations) in the different vegetation types from our transect study. Box plots
show median (red line), interquartile range (IQR) with upper (75%) and lower (25%) quartiles,
lowest whisker still within 1.5IQR of lower quartile, and highest whisker still within 1.5IQR of
upper quartile, dots mark outliers. Abbreviations: con = coniferous forest sites (n=9); dec =
deciduous forest sites (n=14); grass = grassland sites (n=6).

### 3.5 Apparent fractionation of $\delta^2H$ and $\delta^{18}O$ in the different vegetation types

The $\delta^2H$ values could be obtained for $n$-alkanes $C_{27}$, $C_{29}$ and $C_{31}$ in all samples and additionally
at two locations for $n$-$C_{25}$ and $n$-$C_{33}$ at six other locations. The $\delta^2H_{n\text{-alkane}}$ values, calculated as
mean of $n$-$C_{25}$ to $n$-$C_{31}$ $\delta^2H$, ranges from -156 to -216‰. Pooled standard deviations show an
overall average of 3.6‰. The $\delta^{18}O_{sugar}$ values, calculated as the area weighted means for
arabinose and xylose, ranges from 27.7 to 39.4‰. The average weighted mean standard
deviation is 1.4‰. The compound-specific isotope data is summarized along with the
calculations in Tab. S4.





Apparent fractionation ($\varepsilon_{n\text{-alkane/precipitation}}$) is on the order of -120 to -150‰, i.e. a bit less than
the biosynthetic fraction of -160‰. This implies that evapotranspirative enrichment is ~ 10 to
40‰ (Fig. 7A). $\varepsilon_{n\text{-alkane/precipitation}}$ is lower for grass sites compared to the forest sites. Differences
are significant between deciduous and grass sites (p-value = 0.005). This finding supports the
results of other studies (Kahmen et al., 2013; Liu and Yang, 2008; McInerney et al., 2011), and
can be named "signal damping". Grasses do not only incorporate the evaporatively-enriched
leaf water only but also unenriched leaf water in the growth and differentiation zone of grasses
(Gamarra et al., 2016; Liu et al., 2017).
The grass-derived hemicellulose sugar biomarkers do not fully record the evapotranspirative
enrichment of the leaf water, either, as indicated by lower apparent fractionation ($\varepsilon_{\text{sugar/precipitation}}$)
in Fig. 7B. The differences are significant between forest and grass sites (p-value < 0.005). This
is in agreement with a study on cellulose extracted from grass blades (Helliker and Ehleringer,
2002), and again, the "signal damping" can be explained with incorporation of enriched leaf
water and non-enriched stem water.
Based on the comparison of evapotranspirative enrichment between forest and grass sites, the
"signal damping" can be quantified to be ~ 31% for the hemicellulose sugars, and ~ 49% for
the $n$-alkanes. This is in agreement with other studies that reported a loss of 22% of the leaf
water enrichment for hemicellulose sugars (Helliker and Ehleringer, 2002) and 39 to 62% loss
of the leaf water enrichment for $n$-alkanes (Gamarra et al., 2016).

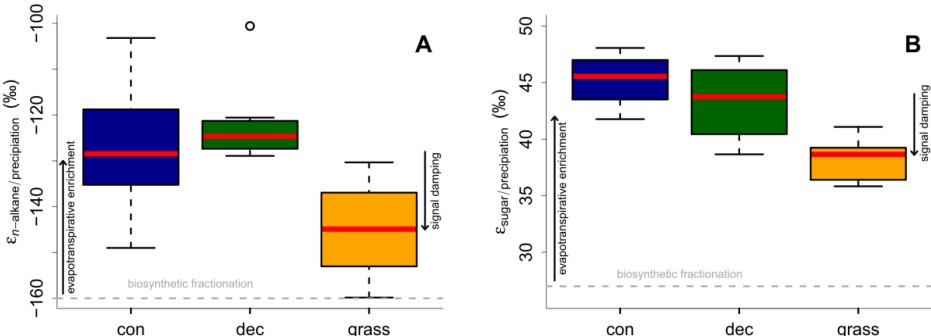


**Fig. 7.** Apparent fractionation (A) $\varepsilon_{n\text{-alkane/precipitation}}$ and (B) $\varepsilon_{\text{sugar/precipitation}}$. Biosynthetic
fractionation factors according to section 2.4.2. Box plots show median (red line), interquartile
range (IQR) with upper (75%) and lower (25%) quartiles, lowest whisker still within 1.5IQR
of lower quartile, and highest whisker still within 1.5IQR of upper quartile, dots mark outliers.
Abbreviations: con = coniferous forest sites (n=9); dec = deciduous forest sites (n=11 and 14
for $n$-alkanes and sugars, respectively); grass = grassland sites (n=4 and 6 for $n$-alkanes and
sugars, respectively). The figure conceptually illustrates the effect of biosynthetic fractionation
and evapotranspirative enrichment as well as "signal damping".

**3.6 $\delta^2$H/$\delta^{18}$O$_{\text{source-water}}$ reconstructions**
The $\delta^2$H versus $\delta^{18}$O diagram shown in Fig. 8 graphically illustrates the reconstruction of
$\delta^2$H$_{n\text{-alkane}}$/$\delta^{18}$O$_{\text{sugar}}$ (crosses), as well as the reconstruction





421 of $\delta^2H/\delta^{18}O_{source-water}$ (black dots). For reconstructing $\delta^2H/\delta^{18}O_{source-water}$, LELs with an average

422 slope of 2.8 ±0.1 (Eq. 10) can be generated through every leaf water point and the intercepts of

423 these LELs with the GMWL.

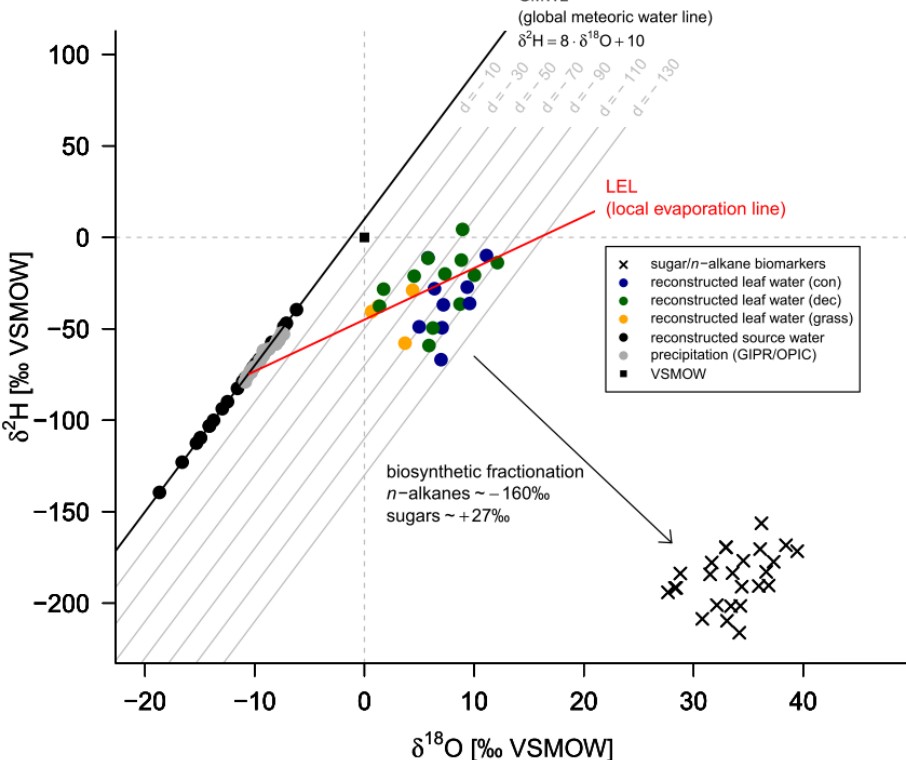


**Fig. 8.** $\delta^2H$ vs. $\delta^{18}O$ diagram illustrating $\delta^2H_{n-alkane}$ and $\delta^{18}O_{sugar}$, reconstructed $\delta^2H/\delta^{18}O_{leaf-water}$ (according Eqs. 8 and 9) and reconstructed $\delta^2H/\delta^{18}O_{source-water}$ in comparison to GIPR/OIPC-based $\delta^2H/\delta^{18}O_{precipitation}$. Abbreviations: con = coniferous forest sites (n=9); dec = deciduous forest sites (n=11); grass = grassland sites (n=4).

429

430 The reconstructed $\delta^2H/\delta^{18}O_{source-water}$ results can be compared with the $\delta^2H/\delta^{18}O_{GIPR/OIPC}$ data

431 (Fig. 9). This comparison reveals that the coupled $\delta^2H_{n-alkane}$-$\delta^{18}O_{sugar}$ approach yields more

432 accurate $\delta^2H/\delta^{18}O_{source-water}$ results than hitherto applied $\delta^2H_{n-alkane}$ single isotope approaches.

433 However, the range of the reconstructed $\delta^2H/\delta^{18}O_{source-water}$ values is clearly larger than in

434 $\delta^2H/\delta^{18}O_{GIPR/OIPC}$ values. $\delta^2H$ is systematically underestimated by ~ 21‰ ±22 (Fig. 9B) and

435 $\delta^{18}O$ by ~ 2.9‰ ±2.8 (Fig. 9D). The type of vegetation seems to be not particularly relevant (p-

436 value = 0.18 for $\Delta\delta^2H$ and p-value = 0.34 for $\Delta\delta^{18}O$). Nevertheless, the systematic offsets tend

437 to be lowest for the decidous sites ($\Delta\delta^2H/\delta^{18}O$ is closer to zero with ~-5‰ ±15 and ~-1.1‰

438 ±2.1), followed by grass sites (~-14‰ ±20 and ~-2.1‰ ±2.6). In comparison, the coniferous

439 sites show the largest offsets (~-23‰ ±26 for $\Delta\delta^2H$ ~-3.0‰ ±3.3 for $\Delta\delta^{18}O$). Differences are,

440 however, not statistically significant. The systematic offset and the large variability might have





more specific reasons, and we suggest that this is related to the type of vegetation. Deciduous
trees produce lots of leaf waxes and sugars (e.g. Prietzel et al., 2013; Zech et al., 2012a), and
all biomarkers reflect and record the evapotranspirative enrichment of the leaf water (e.g.
Cernusak et al., 2016; Tuthorn et al., 2014). The coupled approach and the leaf water
reconstruction based on the $n$-alkane and sugar biomarkers thus works well. However,
coniferous trees produce quite low amounts of $n$-alkanes (Diefendorf and Freimuth, 2016; Zech
et al., 2012a), while sugar concentrations are as high as in other vascular plants (e.g. Hepp et
al., 2016; Prietzel et al., 2013). For the coniferous soil samples this means that the $n$-alkanes
stem most likely from the understory whereas the sugars originate from grasses and coniferous
needles. When the understory is dominated by grass species then the $n$-alkane biomarkers do
not record the full leaf water enrichment signal, whereas the sugars from the needles do. The
reconstructed leaf water for the coniferous sites is therefore too negative concerning $\delta^2H$, and
reconstructed $\delta^2H/\delta^{18}O_{source-water}$ values thus also become too negative (Fig. 8). Concerning the
grass sites the following explanation can be found. Correcting for "signal damping" makes the
reconstructed leaf water points more positive and shifts them in Fig. 8 up and right. As the
"signal damping" is stronger for $\delta^2H$ than for $\delta^{18}O$ the corrected leaf water points are now above
the uncorrected ones. The corrected leaf water points leads to more positive reconstructed
$\delta^2H/\delta^{18}O_{source-water}$ values for the grass sites.
Vegetation type specific rooting depths could partly cause the overall high variability in
reconstructed $\delta^2H/\delta^{18}O_{source-water}$. Deep rooting species most likely use the water from deeper
soil horizons and/or shallow ground water, which is equal to the (weighted) mean annual
precipitation (e.g. Herrmann et al., 1987). Shallow rooting plants take up water from upper soil
horizons, which is influenced by seasonal variations in $\delta^2H/\delta^{18}O_{precipiation}$ and by soil water
enrichment (Dubbert et al., 2013). Thus, the overall assumption that the source water of the
plants reflects the local (weighted) mean precipitation might be not fully valid for all sites.
Moreover, a partly contribution of root-derived rather than leaf-derived sugar biomarkers in our
topsoil samples is very likely. This does, by contrast, not apply for $n$-alkanes, which are hardly
produced in roots (Zech et al., 2012b and the discussion).

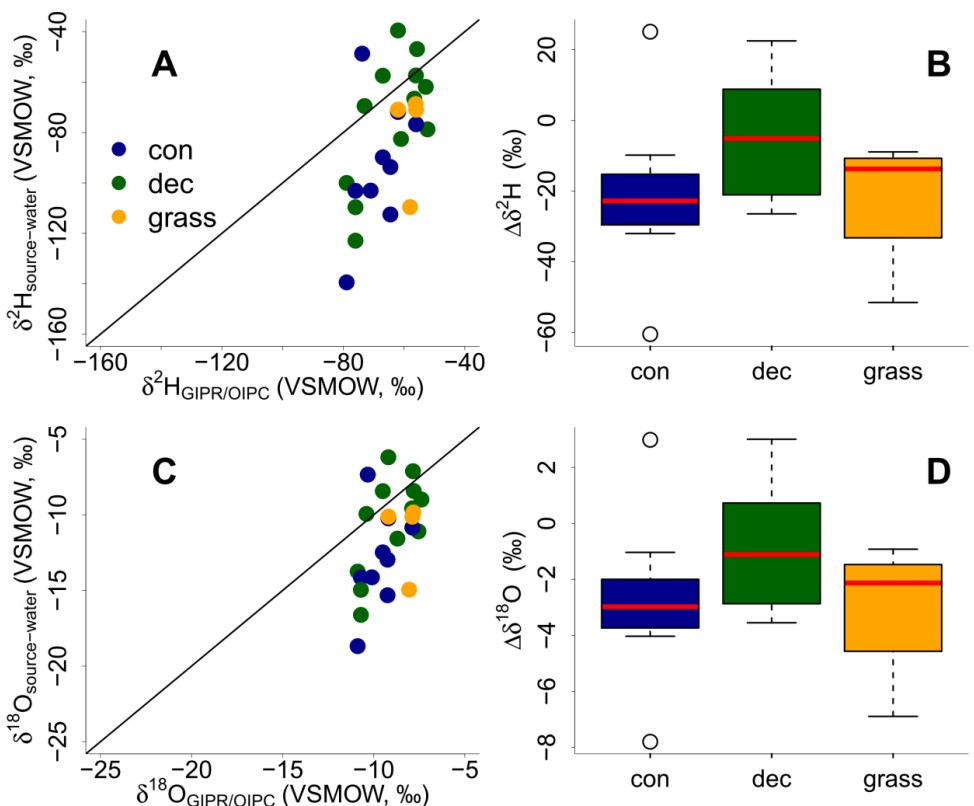

**Fig. 9.** Correlation of reconstructed $\delta^2H/\delta^{18}O_{source-water}$ vs. precipitation $\delta^2H/\delta^{18}O_{GIPR/OIPC}$ (A and C). Black lines indicate 1:1 relationship. Differences between reconstructed source water and precipitation ($\Delta\delta^2H/\delta^{18}O = \delta^2H/\delta^{18}O_{source-water} - \delta^2H/\delta^{18}O_{GIPR/OIPC}$) for the three different vegetation types (B and D). Box plots show median (red line), interquartile range (IQR) with upper (75%) and lower (25%) quartiles, lowest whisker still within 1.5IQR of lower quartile, and highest whisker still within 1.5IQR of upper quartile. Abbreviations: con = coniferous forest sites (n=9); dec = deciduous forest sites (n=11); grass = grassland sites (n=4).

Moreover, the high variability within the vegetation types could be caused by variability in $\varepsilon_{bio}$ of $^2H$ in $n$-alkanes, as well as $^{18}O$ in sugars. There is an ongoing discussion about the correct $\varepsilon_{bio}$ for $^{18}O$ in hemicellulose sugars (Sternberg, 2014 vs. Zech et al., 2014), and $\varepsilon_{bio}$ is probably not constant over all vegetation types. This translates into errors concerning leaf water reconstruction and thus for reconstructing $\delta^2H/\delta^{18}O_{source-water}$ values (Eq. 9 and Fig. 8). Likewise, the $\varepsilon_{bio}$ values reported in the literature for $^2H$ of $n$-alkanes can be off from -160‰ by tens of permille (Feakins and Sessions, 2010; Tipple et al., 2015; Feakins et al., 2016; Freimuth et al., 2017). The degree to which hydrogen originates from NADPH rather than leaf water is important, because NADPH is more negative (Schmidt et al., 2003). The wide range in biosynthetic $^2H$ fractionation factors is therefore also related to the carbon and energy metabolism state of plants (Cormier et al., 2018).



### 3.7 RH reconstruction

Reconstructed $RH_{MDV}$ ranges from 34 to 74%, while $RH_{MDV}$ from climate station data range from 61 to 78% (Fig. 10A). Biomarker-based values thus systematically underestimate the station data ($\Delta RH_{MDV}$ = -17% ±12; Fig. 10B). Yet the offsets are much less for deciduous tree and grass sites ($\Delta RH_{MDV}$ = -10% ±12 and -7% ±9, respectively). The offsets for the coniferous sites are -30% ±11, and significantly larger than for the deciduous and grass sites (p-values < 0.05).

Too low reconstructed $RH_{MDV}$ values for the coniferous sites make sense in view of the previously discussed option that soils contain $n$-alkanes from the understory (which is dominated by grass species), while sugars stem from needles and grasses. As explained earlier already, the "signal damping" leads to too negative reconstructed $\delta^2H_{leaf-water}$ (whereas $\delta^{18}O$ is affected less by the "signal damping"), and too negative $\delta^2H_{leaf-water}$ translates into overestimated d-excess and underestimated RH values. In Fig. 8, a correction for this require moving the coniferous leaf water data points upwards towards more positive $\delta^2H$ values, thus the distance between the leaf water and the source water is reduced.

The underestimation of RH for the deciduous and grass sites could be partly associated with the use of the GMWL as baseline for the coupled $\delta^2H_{n\text{-}alkane}$-$\delta^{18}O_{sugar}$ approach. The deuterium-excess of the LMWLs is generally lower than the +10‰ of the GMWL, while the slopes of the LMWLs are well comparable to the GMWL (Stumpp et al., 2014). In addition, if soil water evaporation occurred before water uptake by the plants, this would lead to an underestimation of biomarker-based $RH_{MDV}$ values. It can be furthermore assumed that plant metabolism is highest during times with direct sunshine and high irradiation, i.e. during noon at sunny days. The relevant RH could therefore be lower than the climate station-derived $RH_{MDV}$. Indeed, already climate station $RH_{MDV}$ is considerable lower than $RH_{MA}$ and $RH_{MV}$ (Tab. S1).

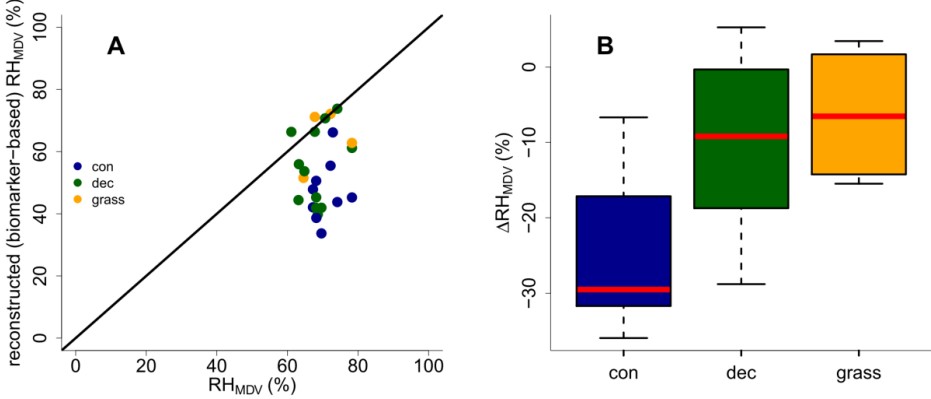

**Fig. 10.** (A) Comparison of reconstructed (biomarker-based) $RH_{MDV}$ values and climate station $RH_{MDV}$ data. The black line indicates the 1:1 relationship. (B) Differences between reconstructed and climate station $RH_{MDV}$ values ($\Delta RH_{MDV}$ = reconstructed − climate station $RH_{MDV}$) for the three different vegetation types along the transect. Abbreviations: con = coniferous forest sites (n=9); dec = deciduous forest sites (n=11); grass = grassland sites (n=4).



The uncertainty of reconstructed $RH_{MDV}$ values are large for all three investigated vegetation
types, and again these uncertainties are probably also related to $\varepsilon_{bio}$, which is most likely not
constant as assumed for our calculations. Moreover, microclimate variability is underestimated
in our approach. As mentioned in sections 2.4.2 and 3.6, in the coupled approach not only the
source water of the plants is equated with (weighted) mean annual precipitation, but also an
isotopic equilibrium between the source water and the (local) atmospheric water vapour is
assumed. However, in areas with distinct seasonality this might be not fully valid. To account
for this lack of equilibrium between precipitation and local atmospheric water vapour, apparent
$\varepsilon$ values can be calculated with data from Jacob and Sonntag, (1991). As shown by Hepp et al.
(2018) those values can be used to achieve alternative RH reconstructions based on the coupled
$\delta^2H_{n\text{-}alkane}$-$\delta^{18}O_{sugar}$ approach. Such calculated $RH_{MDV}$ values are on average 1.5% more
negative than the original values. However, this difference in RH is far below the analytical
uncertainties of the compound-specific biomarker isotope analysis.
Finally, the integration time of the investigated topsoils has to be discussed. Unfortunately, no
$^{14}C$ dates are available for the soil samples. However, most likely the organic matter has been
built up over a longer timescale than the available climate data, which is used for comparison.
In combination with vegetation changes/management changes throughout that period, this
could surely lead to a less tight relationship of the reconstructions compared to the climate
station data. Root input of arabinose and xylose seems to be of minor relevance in our topsoil
samples. Otherwise, the reconstructed $\delta^{18}O_{sugar}$ values would be too negative resulting in
$RH_{MDV}$ overestimations, which is not observed.

## 4 Conclusions

We were able to show that
(i)  the vegetation type does not significantly influence the brGDGT concentrations and
proxies, yet the coniferous sites tend to have higher brGDGT concentrations, BIT
indices and CBT/MBT' ratios, while grass sites tend to be lowest.
(ii)  CBT faithfully records soil pH with a median $\Delta$pH of 0.6 $\pm$0.6, The CBT
overestimates the real pH particularly at the forest sites.
(iii)  CBT-MBT'-derived $T_{MA}$ reflect the climate station-derived $T_{MA}$ values with a
median $\Delta T_{MA}$ of 0.5°C $\pm$2.4, but again slightly too high reconstruction for the forest
sites were observed.
(iv)  differences in the apparent fractionation between the investigated vegetation types
are caused by "signal damping", i.e. the grasses do not see and record the full
evaporative enrichment of leaf water.
(v)  the reconstructed $\delta^2H/\delta^{18}O_{source\text{-}water}$ reflects the $\delta^2H/\delta^{18}O_{GIPR/OIPC}$ with a systematic
offset for $\delta^2H$ of ~-21‰ $\pm$22 and for $\delta^{18}O$ of ~-2.9‰ $\pm$2.8 (based on overall medians
of $\Delta\delta^2H/\delta^{18}O$). This is caused by too negative reconstructions for coniferous and
grass sites. For coniferous sites, this can be explained with $n$-alkanes originating
from understory grasses, and for the grass sites the "signal damping" more effect
$\delta^2H$ than $\delta^{18}O$. This leads to too negative reconstructed $\delta^2H_{leaf\text{-}water}$ values and thus
to too negative $\delta^2H/\delta^{18}O_{source\text{-}water}$ reconstruction.





(vi)     reconstructed (biomarker-based) $RH_{MDV}$ values tend to underestimate climate station-derived $RH_{MDV}$ values ($\Delta RH_{MDV}$ = ~ -17% ±12). For coniferous sites the underestimations are strongest, which can be explained with understory grasses being the main source of *n*-alkanes for the investigated soils under coniferous forests.

Overall, our study highlights the great potential of GDGTs and the coupled $\delta^2H_{\text{n-alkane}}$-$\delta^{18}O_{sugar}$ approach for more quantitative paleoclimate reconstructions. Taking into account effects of different vegetation types improves correlations and reconstructions. This holds particularly true for the coupled $\delta^2H_{\text{n-alkane}}$-$\delta^{18}O_{sugar}$ approach, which is affected by "signal damping" of the grass vegetation. Assuming constant biosynthetic fractionation is likely a considerable source of uncertainty. Climate chamber experiments would be very useful to further evaluate and refine the coupled $\delta^2H_{\text{n-alkane}}$-$\delta^{18}O_{sugar}$ approach, because uncertainties related to microclimate variability can be reduced. Field experiments like ours suffer from the fact that biomarker pools in the sampled topsoils may have been affected by past vegetation and climate changes.

## Acknowledgements

We thank L. Wüthrich, H. Veit, T. Sprafke, A. Groos  (all University of Bern), A. Kühnel (Technical University of Munich) for constructive discussions and statistical advices, and M. Schaarschmidt (University of Bayreuth), C. Heinrich and M. Benesch (Martin-Luther-University Halle-Wittenberg) for laboratory assistance during $\delta^{18}O_{sugar}$ analysis and pH measurements, respectively. The Swiss National Science Foundation (PP00P2 150590) funded this research. J. Hepp greatly acknowledges the support by the German Federal Environmental Foundation (DBU) in form of his PhD-fellowship.

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
