# Peer review of "Evaluation of bacterial glycerol dialkyl glycerol tetraether and $^2$H-$^{18}$O biomarker proxies along a Central European topsoil transect"

_Biogeosciences, 2019_

## Referee Comment (RC1) · Anonymous Referee #1 · 8 Jul 2019

Hepp et al. measured branched GDGT concentrations, hydrogen isotopes of leaf waxes, and oxygen isotopes of sugars from surface soils collected throughout Germany and Sweden. They use their measurements to add to existing calibrations of brGDGT proxies for soil pH and temperature, and to test a proposed method to reconstruct relative humidity from paired $\delta2H$ values of n-alkanes and $\delta18O$ values of sugars. While I agree that improving calibrations of paleoclimate proxies is important for interpreting them, I have some major concerns about the utility of the present manuscript. Unfortunately, I do not think the study is suitable for publication in Biogeosciences as presently written. With some major restructuring and rewriting, I think the authors could still publish something useful from their data set.

[Figure]

Major issues: 1) The brGDGT calibration presented here is of limited use, since the study uses an outdated method to measure brGDGTs and does not distinguish between the 5 methyl and 6 methyl compounds. Hepp et al. thus calibrate indices (CBT and MBT') that have fallen out of favor and been replaced by the more robust CBT' and MBT5Me indices. The new indices and new methods developed by De Jonge et al. (GCA, 2014, doi: 10.1016/j.gca.2014.06.013) and Hopmans et al. (Organic Geochemistry, 2016, doi: 10.1016/j.orggeochem.2015.12.006) are not even mentioned in the text, and the limitations of the brGDGT data presented here are not acknowledged. Without reanalyzing these samples with a method that resolves all isomers, I fear that the present sample set has limited value for the calibration of brGDGT-based proxies.

2) There are some big assumptions in the proposed approach for reconstructing relative humidity using paired $\delta 2H$ values of n-alkanes and $\delta 18O$ values of sugars. In particular, the assumption that biosynthetic fractionation for these compounds is constant is contradicted by lots of existing work, which is briefly mentioned by the authors in their discussion. Figure 8 is not a very good advertisement for the utility of the paired $\delta 2H$-alkane/$\delta 18O$ sugar approach, and the lack of correlation suggests that some of the many assumptions that go into this method are not valid. This paired approach has not caught on beyond the Zech group, and the data presented here suggests that it may not be useful as presently conceived. The authors state they have shown the "great potential" for this proxy. I remain unconvinced by the data and analysis shown here.

3) The writing is in places unclear and difficult to follow. I have noted a few of these instances in my technical corrections, but the manuscript would benefit from more careful editing.

Specific comments:

Line 110: This adds up to more than 16, some sites were considered to be more than one of these categories? Would be good to rewrite to clarify

Line 114: Was there a threshold for what was considered "close-by"?

Line 133: Machine learning techniques like random forest aren't so commonly used in Biogeosciences and it would be helpful to provide more details here. How many trees did you use? How was data partitioned into training and testing sets? What metric was used to assess model performance? What was the minimum number of samples in the terminal nodes? What was the maximum number of terminal nodes? What variables ended up being ranked as most important (could be useful to show a plot of ranked variable importance in the supplemental materials)?

Line 136: Why wasn't it possible? Lack of measured data for a robust training data set? Please specify

Lines 128-139: How did the calculated values you obtained for the German sites compare to OIPC? What is your evidence for your approach providing superior estimates of precip isotopes than OIPC? OIPC is obviously not perfect, but as written, we have no evidence to evaluate if your results are any more accurate. There is also no discussion of the implications of using one target for precip isotopes in the southern half of your transect and a different one in the northern half.

Section 2.3.1: No internal standard was added? How do you account for losses of brGDGTs during sample handling?

Lines 165-171: This is not the most current method used for robust brGDGT analysis (see Hopmans et al., Organic Geochemistry, 2016. DOI: 10.1016/j.orggeochem.2015.12.006). Does your method allow for 5' and 6' methyl brGDGTs to be distinguished from one another? If not, severely diminishes the accuracy of results. Based on the results that are shown, it seems like this method does not distinguish the different isomers.

Lines 172-173: how was the pH measured?

Section 2.3.2: Were the n-alkanes quantified prior to measuring their stable isotopes? Also, please briefly describe the operating conditions of the GC-pyr-IRMS (or cite another publication that used an identical method and provides all the relevant details)

Line 199: It is not clear how you had 29 samples from 16 sites. Were some of the sites sampled in duplicate?

Lines 211-221: The more robust indicator of soil pH is CBT' and the more robust indicator of soil temperature is MBT5me (De Jonge et al., GCA, 2014, DOI:10.1016/j.gca.2014.06.013).

Lines 227-229: A number of papers have shown that ebio is not constant and different among plant types and seasonally. See for example Feakins & Sessions 2010 (cited previously), Eley et al., GCA, 2014 (DOI: 10.1016/j.gca.2013.11.045), Cormier et al., New Phytologist, 2018 (DOI: 10.1111/nph.15016).

Lines 383-385: are these concentrated weighted means? That is what is typically used to compare d2H values of n-alkanes where not all homologues are present in all samples

Line 395: I think you mean "unenriched xylem water"?

Lines 431-432: This is not particularly convincing, the reconstructed precipitation isotopes are not correlated with the GIPR/OIPC precipitation isotopes. No evidence is provided to show that this approach is any better than the most up to date methods for obtaining precipitation isotopes from leaf wax n-alkane isotopes alone. For example, how do your results compare to the predictions from the proxy system model developed by Konecky et al. (JGR-Biogeoscience, 2019, DOI: 10.1029/2018JG004708)? Maybe your approach is better, but you need to prove this by providing a direct comparison, rather than just telling us

Lines 448-450: If this was the case, wouldn't you expect all the coniferous sites to be biased in the same direction? Instead, they are evenly distributed above and below the 1:1 line

Line 454: Is this signal damping correction shown anywhere? How would this work

[Figure]

practically in sediments?

Lines 467-468: Actually, there are plenty of n-alkanes in roots and they have very different H isotopic composition than in leaves. See work from Guido Wiesenberg's group and Gamarra and Kahmen. I'm also confused about what you are referring to as "the discussion". There is not a separate discussion section to this manuscript.

Lines 489-494: Not stated here is that there is no correlation between the reconstructed and measured RH values. This suggests that this approach for reconstructing RH is not particularly useful Line 565: The data in the paper is not very convincing that there is great potential for the coupled d2H n-alkane d18O sugar approach

Lines 566-567: I don't see evidence of this in your analysis, nor examples of how you would take vegetation into account when applying this proxy.

Technical corrections and typing errors:

Lines 54-56: The way this sentence is written is confusing. Suggest rewriting as "Climate proxies based on molecular fossils, also known as biomarkers, have great potential...

Line 56: don't need the comma after "particular"

Line 59: "need to be known"?

Line 61: It would be better to start this paragraph with a clear link back to the previous one

Line 74: don't need commas before and after "it is known"

Line 79: Again, some sort of transition would be helpful to begin this paragraph

Line 82: "all along the way" too wordy

Lines 93-94: "as well as concerning possible effects related to" awkward phrasing

Figure 1: would be nice to have a legend on panel B or have the axis colors match the

variable colors. At the moment we are left to guess that blue bars are precip and the red dots are temp, since this is not stated in the figure caption or the legend.

Also would be nice to offset the panel letters with a () or . to break them apart form the title of the panel

Line 180: No "the" needed in front of ETH

Line 225: the n at the beginning of n-alkane should be italicized. Check throughout

Line 234: Generally, figures should be numbered in the same order that they are referenced in the text

---

## Referee Comment (RC2) · Anonymous Referee #2 · 14 Aug 2019

GENERAL:

The topic of the manuscript is interesting and important as it deals with the evaluation of highly promising proxies used to reconstruct past environmental conditions. While the data produced are rare and are certainly worth publishing, the manuscript has severe flaws that prevent, in my opinion, its publication in this form.

MAJOR PROBLEMS:

A) While reading the manuscript, the connection between GDGT and the plant proxies (i.e. n-alkanes and hemicellulose) is not clear and seems disconnected as if from two separate manuscripts. Moreover, in the section 3.1 of the discussion, the GDGT

data are presented in a way leading the readers to believe that these molecules are produced by plants.

B) The other major point is that the authors suggest that it is "often" not feasible to disentangle between the evapotranspirative enrichment from the precipitation signal, but there is at least another well-established method to do so and published in Climate of the Past (see recent Sachse's group publications, e.g. A dual-biomarker approach for quantification of changes in relative humidity from sedimentary lipid D/H ratios, Climate of the Past, 2017). While this method should at least be mentioned, I also believe the method should be compared to help the readers understand the full set of tools available to study that issue. These two methods are very likely to be highly complementary.

SPECIFICS:

Line 298 to 303: This section is not clear due to some typos or mistakes, please reformulate.

Line 389 to 407: While the difference of ebio is reported at the end of the section (around line 477 to 487), the possibility that a variable ebio could explain the different signals in different types of vegetation, beside the damping effect, is evacuated of the discussion. This should at least be discussed.

Line 432: Is that referring to simply using isotope values of a single compound? What is that hitherto method (reference missing?)? I believe this brings us back to the problem B. The results would gain a lot to be compared with the updated tool box of proxies.

Line 444 to 458: The argumentation is not clear/convincing, please reformulate.

Line 483-484: The idea of a variable ebio is well expressed in general, but references to some recent works is missing that shows even greater variability in n-alkane dD values under different metabolisms (e.g. Cormier et al, 2018 – New Phytologist, Tipple & Ehleringer 2018 – Oecologia, Cormier et al, 2019 – Oecologia)

[Figure]

Line 490 to 494: Please reformulate, this section is not clear.

Line 550: If the author are really considering a variable ebio, the damping effect can only potentially explain the different signals observed in different types of vegetation. Again, ebio should be part of the points because standing alone, they can induce confusion even if mentioned afterward.

---

## Author Comment (AC1) · 4 Sep 2019

**Reply to Referee #2**

by Johannes Hepp, Michael and Roland Zech & co-authors

*GENERAL:*

*The topic of the manuscript is interesting and important as it deals with the evaluation of highly promising proxies used to reconstruct past environmental conditions. While the data produced are rare and are certainly worth publishing, the manuscript has severe flaws that prevent, in my opinion, its publication in this form.*

→ While we are grateful to Referee #2 for her/his constructive suggestions helping to improve our manuscript, the two raised 'major problems' are certainly no 'severe flaws' preventing publication (see our replies below). We therefore see no justification for rejection of our manuscript based on the review provided by Referee #2.

*MAJOR PROBLEMS:*

*A) While reading the manuscript, the connection between GDGT and the plant proxies (i.e. n-alkanes and hemicellulose) is not clear and seems disconnected as if from two separate manuscripts. Moreover, in the section 3.1 of the discussion, the GDGT data are presented in a way leading the readers to believe that these molecules are produced by plants.*

→ We are very surprised that Referee #2 considers the GDGT and the $n$-alkane/sugar biomarker approach as disconnected. We disagree. Both approaches are based on biomarkers/molecular proxies and are used for paleoclimate reconstructions. We clearly state and explain in the introduction and method sections how the applied biomarkers (GDGT´s as well as $n$-alkanes and sugars) are produced, how calculations are done and how the proxies can be interpreted. Please note that there are plenty of studies in the literature presenting both GDGT and $\delta^2 H_{n\text{-alkane}}$ results in one publication → **certainly no major problem/severe flaw**.

*B) The other major point is that the authors suggest that it is "often" not feasible to disentangle between the evapotranspirative enrichment from the precipitation signal, but there is at least another well-established method to do so and published in Climate of the Past (see recent Sachse's group publications, e.g. A dual-biomarker approach for quantification of changes in relative humidity from sedimentary lipid D/H ratios, Climate of the Past, 2017). While this method should at least be mentioned, I also believe the method should be compared to help the readers understand the full set of tools available to study that issue. These two methods are very likely to be highly complementary.*

→ Please note that the 'dual biomarker approach' of Rach et al. (2017, CP) is not applicable to terrestrial (soil) samples/archives, but only to lacustrine settings. A comparison with our 'coupled $\delta^2 H_{n\text{-alkane}}$-$\delta^{18} O_{sugar}$ biomarker approach' is therefore neither possible nor reasonably within our European climate transect study → **certainly no major problem/severe flaw**.

For a critical evaluation and assessment of both approaches when applied to lacustrine paleoclimate archives, we kindly refer our readers to Hepp et al. (2019, CP) and to our 'Reply to D. Sachse and F. Schenk (SC4: Data analysis and paleoclimatic context)' available online via https://www.clim-past-discuss.net/cp-2018-114/cp-2018-114-AC6-supplement.pdf. Accordingly, the major shortcomings/uncertainties of the 'dual biomarker approach' of Rach et al. (2017) are (i) biosynthetic fraction, (ii) the assumption that paleo-lake water is not affected by evaporative enrichment and (iii) the assumption that the alkane $nC_{23}$ in lacustrine sediments is of aquatic origin. At least for Central European case studies, the latter assumption is certainly not valid, because birch produces considerable amounts of mid-chain $n$-alkanes such as $nC_{23}$. This is acknowledged e.g. by Aichner et al. (2018, CP) concluding for a palaeo lake from Poland that "…mid-chain compounds, which are often interpreted as of aquatic origin, are here rather a mixture of aquatic and terrestrial sources, with high proportional input of the latter during certain time periods." → This short excursion highlights the need for alternative approaches and justifies the testing/evaluation of our 'coupled $\delta^2H_{n\text{-alkane}}$-$\delta^{18}O_{sugar}$ biomarker approach' as it was done by Tuthorn et al. (2015, BG) for an Argentinian climate transect and as is done in the here presented European climate transect study.

*SPECIFICS:*

*Line 298 to 303: This section is not clear due to some typos or mistakes, please reformulate.*

→ Changed.

*Line 389 to 407: While the difference of ebio is reported at the end of the section (around line 477 to 487), the possibility that a variable ebio could explain the different signals in different types of vegetation, beside the damping effect, is evacuated of the discussion. This should at least be discussed.*

→ Changed.

*Line 432: Is that referring to simply using isotope values of a single compound? What is that hitherto method (reference missing?)? I believe this brings us back to the problem B. The results would gain a lot to be compared with the updated tool box of proxies.*

→ The sentence was slightly changed. See also our reply to 'major problem B'.

*Line 444 to 458: The argumentation is not clear/convincing, please reformulate.*

→ We deleted the respective sentence from the revised version of the manuscript.

*Line 483-484: The idea of a variable ebio is well expressed in general, but references to some recent works is missing that shows even greater variability in n-alkane dD values under different metabolisms (e.g. Cormier et al, 2018 – New Phytologist, Tipple & Ehleringer 2018 – Oecologia, Cormier et al, 2019 – Oecologia)*

→ Please note that we already included Cormier et al. (2018) in the actual version of the manuscript and that the fact is mentioned that $\varepsilon_{bio}$ can range even larger when also the metabolic status of the plants is considered. However, we changed the respective sentence to:

"The wide range in biosynthetic $^2$H fractionation factors, which can be even larger, is therefore also related to the carbon and energy metabolism state of plants (Cormier et al., 2018).".

*Line 490 to 494: Please reformulate, this section is not clear.*

→ We changed the quoting of Fig. 10B.

*Line 550: If the author are really considering a variable ebio, the damping effect can only potentially explain the different signals observed in different types of vegetation. Again, ebio should be part of the points because standing alone, they can induce confusion even if mentioned afterward.*

→ You are right. Gao et al. (2014) and Liu et al. (2016) showed that the $\varepsilon_{bio}$ of monocot plants could larger than those of dicots. This would therefore course a more negative apparent fractionation factor for grasses compared to trees. We observe that the apparent fractionation is indeed more negative for the grass sites compared to the forest sites. We will included a discussion about the indistinguishable effects of "signal damping" vs. variable $\varepsilon_{bio}$ along with vegetation types in the respective parts of the manuscript.

**References**

Aichner, B., Ott, F., Słowiński, M., Noryśkiewicz, A. M., Brauer, A. and Sachse, D.: Leaf wax *n*-alkane distributions record ecological changes during the Younger Dryas at Trzechowskie paleolake (Northern Poland) without temporal delay, Climate of the Past Discussions, (March), 1–29, doi:10.5194/cp-2018-6, 2018.

Cormier, M.-A., Werner, R. A., Sauer, P. E., Gröcke, D. R., M.C., L., Wieloch, T., Schleucher, J. and Kahmen, A.: $^2$H fractiontions during the biosynthesis of carbohydrates and lipids imprint a metabolic signal on the $\delta^2$H values of plant organic compounds, New Phytologist, 218(2), 479–491, doi:10.1111/nph.15016, 2018.

Gao, L., Edwards, E. J., Zeng, Y. and Huang, Y.: Major evolutionary trends in hydrogen isotope fractionation of vascular plant leaf waxes, PLoS ONE, 9(11), doi:10.1371/journal.pone.0112610, 2014.

Hepp, J., Wüthrich, L., Bromm, T., Bliedtner, M., Schäfer, I. K., Glaser, B., Rozanski, K., Sirocko, F., Zech, R. and Zech, M.: How dry was the Younger Dryas? Evidence from a coupled $\delta^2$H–$\delta^{18}$O biomarker paleohygrometer applied to the Gemündener Maar sediments, Western Eifel, Germany, Climate of the Past, 15, 713–733, doi:10.5194/cp-15-713-2019, 2019.

Liu, J., Liu, W., An, Z. and Yang, H.: Different hydrogen isotope fractionations during lipid formation in higher plants: Implications for paleohydrology reconstruction at a global scale, Scientific Reports, 6, 19711, doi:10.1038/srep19711, 2016.

Rach, O., Kahmen, A., Brauer, A. and Sachse, D.: A dual-biomarker approach for quantification of changes in relative humidity from sedimentary lipid D/H ratios, Climate of the Past, 13, 741–757, doi:10.5194/cp-2017-7, 2017.

Tuthorn, M., Zech, R., Ruppenthal, M., Oelmann, Y., Kahmen, A., del Valle, H. F., Eglinton, T.,

Rozanski, K. and Zech, M.: Coupling $\delta^2$H and $\delta^{18}$O biomarker results yields information on relative humidity and isotopic composition of precipitation - a climate transect validation study, Biogeosciences, 12, 3913–3924, doi:10.5194/bg-12-3913-2015, 2015.

---

## Author Comment (AC2) · 4 Sep 2019

**Reply to Referee #1**

by Johannes Hepp, Michael and Roland Zech & co-authors

We are grateful to anonymous Referee #1 for her/his constructive suggestions helping to improve our manuscript. Please find our replies to the individual comments below.

*Major issues:*

*1) The brGDGT calibration presented here is of limited use, since the study uses an outdated method to measure brGDGTs and does not distinguish between the 5 methyl and 6 methyl compounds. Hepp et al. thus calibrate indices (CBT and MBT') that have fallen out of favor and been replaced by the more robust CBT' and MBT5Me indices. The new indices and new methods developed by De Jonge et al. (GCA, 2014, doi: 10.1016/j.gca.2014.06.013) and Hopmans et al. (Organic Geochemistry, 2016, doi: 10.1016/j.orggeochem.2015.12.006) are not even mentioned in the text, and the limitations of the brGDGT data presented here are not acknowledged. Without reanalyzing these samples with a method that resolves all isomers, I fear that the present sample set has limited value for the calibration of brGDGT-based proxies.*

→ Referee #1 is right in his/her statement that the GDGT data presented in our manuscript were not acquired based on the up-to-date method. During revision, we will therefore explicitly emphasize that meanwhile new indices and methods were developed (including citations recommended by Referee #1). We would still see a high value of having our GDGT dataset published, because our results fit well to the calibrations done with the previous approach and this in turn allows evaluating GDGT proxy data published for Europe based on the previous approach.

*2) There are some big assumptions in the proposed approach for reconstructing relative humidity using paired δ2H values of n-alkanes and δ18O values of sugars. In particular, the assumption that biosynthetic fractionation for these compounds is constant is contradicted by lots of existing work, which is briefly mentioned by the authors in their discussion. Figure 8 is not a very good advertisement for the utility of the paired δ2H-alkane/δ18O sugar approach, and the lack of correlation suggests that some of the many assumptions that go into this method are not valid. This paired approach has not caught on beyond the Zech group, and the data presented here suggests that it may not be useful as presently conceived. The authors state they have shown the "great potential" for this proxy. I remain unconvinced by the data and analysis shown here.*

→ We accept that Referee #1 remains unconvinced by our coupled $\delta^2H_{n\text{-alkane}}$-$\delta^{18}O_{sugar}$ biomarker approach. We moreover (i) agree, (ii) are aware and (iii) explicitly state that the assumption of constant biosynthetic fraction is likely a major uncertainty of our approach. Still we are convinced that the **'opening of the second dimension'** by our group is a cutting-edge step forward and more promising than focusing on $\delta^2H_{n\text{-alkane}}$ alone. The reason for other working groups not having caught on the coupled approach might have to be seen, in our opinion, in the uniqueness of compound-specific $\delta^{18}O$ analyses: according to our knowledge, only 3 working groups world-wide have respective experience/publication records. Still, we

would be delighted to see the coupled approach being tested or applied by other groups, readily in cooperation with us. Please also note and be aware that any $\delta^2H_{n\text{-alkane}}$ paleoclimate study (without $^2H$-$^{18}O$ coupling!) could be rejected arguing with the uncertainty of biosynthetic fractionation, too.

Possibly, Referee #1 misunderstood Fig. 8. No correlation for the data points shown in Fig. 8 are to be expected. We clarified in our revision that Fig. 8 illustrates the 'concept of the coupled $\delta^2H_{n\text{-alkane}}$-$\delta^{18}O_{sugar}$ biomarker approach'. This conceptual figure illustrates (together with Fig. 9) that $\delta^2H/\delta^{18}O_{prec}$ values reconstructed by the coupled approach are more accurate than $\delta^2H_{prec}$ values reconstructed using $\delta^2H_{n\text{-alkane}}$ alone. Moreover, Fig. 10 illustrates that reconstructed RH values under deciduous forest sites and grassland sites are quite well in accordance with RH values of climate stations, thus indeed demonstrating the great potential of the coupled approach.

*3) The writing is in places unclear and difficult to follow. I have noted a few of these instances in my technical corrections, but the manuscript would benefit from more careful editing.*

→ We will insure a technical and grammatical improvement for the revised version of the manuscript.

*Specific comments:*

*Line 110: This adds up to more than 16, some sites were considered to be more than one of these categories? Would be good to rewrite to clarify*

→ Following the recommendation of Referee #1 we will restructure this sentence. The revised version will read: "In November 2012, we collected 29 topsoil samples (0-5 cm depth) from 16 sites along a transect from Southern Germany to Southern Sweden (Fig. 1A). We distinguished between coniferous forest (con, n = 9), …".

*Line 114: Was there a threshold for what was considered "close-by"?*

→ We agree with Referee #1 that this was not obvious so far in the manuscript and especially not in the supplementary material where the longitude, latitude and altitude were provided for the climate stations (Tab. S2) but not for the locations/sites. In the revised manuscript, we will add the respective characteristics to Tab S1.

*Line 133: Machine learning techniques like random forest aren't so commonly used in Biogeosciences and it would be helpful to provide more details here. How many trees did you use? How was data partitioned into training and testing sets? What metric was used to assess model performance? What was the minimum number of samples in the terminal nodes? What was the maximum number of terminal nodes? What variables ended up being ranked as most important (could be useful to show a plot of ranked variable importance in the supplemental materials)?*

→ As suggested, we will add a supplementary method description part and refer to it in the text.

*Line 136: Why wasn't it possible? Lack of measured data for a robust training data set? Please specify*

→ Because no precipitation isotope data was available for the Danish and Swedish sites.

*Lines 128-139: How did the calculated values you obtained for the German sites compare to OIPC? What is your evidence for your approach providing superior estimates of precip isotopes than OIPC? OIPC is obviously not perfect, but as written, we have no evidence to evaluate if your results are any more accurate. There is also no discussion of the implications of using one target for precip isotopes in the southern half of your transect and a different one in the northern half.*

→ Please allow us to refer to the (cited) Diploma Thesis of Schlotter (2007): there are numerous reasons mentioned already in the introduction highlighting that OPIC is probably not the most robust estimator for middle and high latitudes. That's why we used our own regionalization where it was possible.

*Section 2.3.1: No internal standard was added? How do you account for losses of brGDGTs during sample handling?*

→ We used standard laboratory procedure for GDGT sample preparation. The internal standard was, as written, added before the measurements. A correction for GDGT losses during sample preparation is therefore not possible.

*Lines 165-171: This is not the most current method used for robust brGDGT analysis (see Hopmans et al., Organic Geochemistry, 2016. DOI: 10.1016/j.orggeochem.2015.12.006). Does your method allow for 5' and 6' methyl brGDGTs to be distinguished from one another? If not, severely diminishes the accuracy of results. Based on the results that are shown, it seems like this method does not distinguish the different isomers.*

→ That's correct. Please see our reply to major issue 1.

*Lines 172-173: how was the pH measured?*

→ We will include the information that a pH meter was used.

*Section 2.3.2: Were the n-alkanes quantified prior to measuring their stable isotopes?*

→ Yes, namely by Schäfer et al. (2016). We therefore added the following sentence in the section: "For more details about *n*-alkane quantification the reader is referred to Schäfer et al. (2016). ".

*Also, please briefly describe the operating conditions of the GC-pyr-IRMS (or cite another publication that used an identical method and provides all the relevant details)*

→ As suggested, we added now in the revised version of the manuscript a reference (Christoph et al., 2019), in which the method is described in more detail and we added that the $^2$H pyrolysis reactor temperature was kept at 1420 °C.

*Line 199: It is not clear how you had 29 samples from 16 sites. Were some of the sites sampled in duplicate?*

→ We will clarify during revision that 29 samples were collected from 16 sites. These are, however, no duplicates, but rather different dominant vegetation types (see reply above).

*Lines 211-221: The more robust indicator of soil pH is CBT' and the more robust indicator of soil temperature is MBT5me (De Jonge et al., GCA, 2014, DOI:10.1016/j.gca.2014.06.013).*

→ See our reply to major issue 1.

*Lines 227-229: A number of papers have shown that ebio is not constant and different among plant types and seasonally. See for example Feakins & Sessions 2010 (cited previously), Eley et al., GCA, 2014 (DOI: 10.1016/j.gca.2013.11.045), Cormier et al., New Phytologist, 2018 (DOI: 10.1111/nph.15016).*

→ That's true and especially important when only $\delta^2H_{n\text{-alkane}}$ is used to reconstruct $\delta^2H_{leaf\text{-water}}$. Nevertheless, we emphasize in our manuscript that $\epsilon_{bio}$ is a major uncertainty in our coupled approach, too. At the same time, it's exactly such uncertainties why we need climate transect calibration studies as the one presented here for Europe.

*Lines 383-385: are these concentrated weighted means? That is what is typically used to compare d2H values of n-alkanes where not all homologues are present in all samples*

→ We used here mean values, because the areas and concentrations where not determined during isotope measurements.

*Line 395: I think you mean "unenriched xylem water"?*

→ Yes, changed.

*Lines 431-432: This is not particularly convincing, the reconstructed precipitation isotopes are not correlated with the GIPR/OIPC precipitation isotopes. No evidence is provided to show that this approach is any better than the most up to date methods for obtaining precipitation isotopes from leaf wax n-alkane isotopes alone. For example, how do your results compare to the predictions from the proxy system model developed by Konecky et al. (JGR-Biogeoscience, 2019, DOI: 10.1029/2018JG004708)? Maybe your approach is better, but you need to prove this by providing a direct comparison, rather than just telling us*

→ Please note that we do not necessarily expect a good correlation of our reconstructed $\delta^2H/\delta^{18}O_{prec}$ values with the GIPR/OIPC data, but rather a good (accurate) match on the 1:1 line. Nevertheless, many thanks for pointing us to the new publication by Konecky et al. (2019). While we will readily include a respective citation, we think that a direct comparison of our approach with the one suggested by Konecky et al. (2019) would be beyond the scope of our manuscript.

*Lines 448-450: If this was the case, wouldn't you expect all the coniferous sites to be biased in the same direction? Instead, they are evenly distributed above and below the 1:1 line*

→ No, please see Fig. 9: we do not see that the coniferous sites are evenly distributed around the 1:1 line. Except for one data point, they are clearly below the 1:1 line.

*Line 454: Is this signal damping correction shown anywhere? How would this work practically in sediments?*

→ No, sorry. This signal damping correction is not shown or quantified in this manuscript. This would require a quantitative estimation of the contribution of grass vegetation to the total biomass pool in the topsoil. For an example how such a correction can be applied to lake sediments please see e.g. Hepp et al. (2019, CP).

*Lines 467-468: Actually, there are plenty of n-alkanes in roots and they have very different H isotopic composition than in leaves. See work from Guido Wiesenberg's group and Gamarra and Kahmen. I'm also confused about what you are referring to as "the discussion". There is not a separate discussion section to this manuscript.*

→ Changed to "Zech et al. ,2012b and the discussion therein". We do not agree and we are not aware of any new studies showing that *n*-alkanes are produced in large amounts by roots in comparison to leaves. There is a clear respective dissense with the Wiesenberg group (see open discussion of Zech et al., 2012b, or Zech et al., 2013 - Response to the comments by G. Wiesenberg and M. Gocke. Quaternary Research 79(2), 306-307). Moreover, the work of Gamarra and Kahmen (2015) shows that root *n*-alkane concentration is always the lowest compared to the other plant tissues sampled.

*Lines 489-494: Not stated here is that there is no correlation between the reconstructed and measured RH values. This suggests that this approach for reconstructing RH is not particularly useful Line 565: The data in the paper is not very convincing that there is great potential for the coupled d2H n-alkane d18O sugar approach*

→ We don't agree. Given the low range of measured RH values along this European climate transect and the uncertainties of the coupled approach for reconstructing RH values, the lack of a respective correlation is not really surprising to us. Please compare a similar climate transect study by Tuthorn et al. (2015, BG) where the RH range is much larger and where indeed a significant correlation can be found. For this European transect study here, the usefulness of the coupled approach for reconstructing RH values should be rather inferred from the quite well 1:1 match for deciduous forest sites and grassland sites (cf. Fig. 9). The RH underestimation for coniferous forest sites can be easily explained with the extremely low n-alkane production of coniferous trees (see ll. 495-502).

*Lines 566-567: I don't see evidence of this in your analysis, nor examples of how you would take vegetation into account when applying this proxy.*

→ See for example Hepp et al. (2019).

*Technical corrections and typing errors:*

*Lines 54-56: The way this sentence is written is confusing. Suggest rewriting as "Climate proxies based on molecular fossils, also known as biomarkers, have great potential...*

→ Changed.

*Line 56: don't need the comma after "particular"*

→ Changed.

*Line 59: "need to be known"?*

→ Changed.

*Line 61: It would be better to start this paragraph with a clear link back to the previous one*

→ We now start the paragraph with "One famous and widely applied lipid biomarker group are terrestrial branched glycerol dialkyl glycerol tetraethers (brGDGTs). They are synthesized… and…"

*Line 74: don't need commas before and after "it is known"*

→ Changed.

*Line 79: Again, some sort of transition would be helpful to begin this paragraph*

→ We now start the paragraph with "Concerning paleohydrology proxies, compound-specific…"

*Line 82: "all along the way" too wordy*

→ Changed

*Lines 93-94: "as well as concerning possible effects related to" awkward phrasing*

→ Changed.

*Figure 1: would be nice to have a legend on panel B or have the axis colors match the variable colors. At the moment we are left to guess that blue bars are precip and the red dots are temp, since this is not stated in the figure caption or the legend. Also would be nice to offset the panel letters with a () or . to break them apart from the title of the panel*

→ Changed.

*Line 180: No "the" needed in front of ETH*

→ Changed.

*Line 225: the n at the beginning of n-alkane should be italicized. Check throughout*

→ Changed.

*Line 234: Generally, figures should be numbered in the same order that they are referenced in the text*

→ Checked and changed if necessary.

**References**

Christoph, H., Eglinton, T. I., Zech, W., Sosin, P. and Zech, R.: A 250 ka leaf-wax δD record from a loess section in Darai Kalon , Southern Tajikistan, Quaternary Science Reviews, 208, 118–128, doi:10.1016/j.quascirev.2019.01.019, 2019.

Gamarra, B. and Kahmen, A.: Concentrations and $\delta^2$H values of cuticular *n*-alkanes vary significantly among plant organs, species and habitats in grasses from an alpine and a temperate European grassland, Oecologia, 178, 981–998, doi:10.1007/s00442-015-3278-6, 2015.

Hepp, J., Wüthrich, L., Bromm, T., Bliedtner, M., Schäfer, I. K., Glaser, B., Rozanski, K., Sirocko, F., Zech, R. and Zech, M.: How dry was the Younger Dryas? Evidence from a coupled $\delta^2$H–$\delta^{18}$O biomarker paleohygrometer applied to the Gemündener Maar sediments, Western Eifel, Germany, Climate of the Past, 15, 713–733, doi:10.5194/cp-15-713-2019, 2019.

Konecky, B., Dee, S. G. and Noone, D. C.: WaxPSM: A Forward Model of Leaf Wax Hydrogen Isotope Ratios to Bridge Proxy and Model Estimates of Past Climate, Journal of Geophysical Research: Biogeosciences, 124, 2107–2125, doi:10.1029/2018JG004708, 2019.

Schlotter, D.: The spatio-temporal distribution of $\delta^{18}$O and $\delta^2$H of precipitation in Germany - an evaluation of regionalization methods, Albert-Ludwigs-Universität Freiburg im Breisgau. [online] Available from: http://www.hydrology.uni-freiburg.de/abschluss/Schlotter_D_2007_DA.pdf, 2007.

Tuthorn, M., Zech, R., Ruppenthal, M., Oelmann, Y., Kahmen, A., del Valle, H. F., Eglinton, T., Rozanski, K. and Zech, M.: Coupling $\delta^2$H and $\delta^{18}$O biomarker results yields information on relative humidity and isotopic composition of precipitation - a climate transect validation study, Biogeosciences, 12, 3913–3924, doi:10.5194/bg-12-3913-2015, 2015.

Zech, M., Kreutzer, S., Goslar, T., Meszner, S., Krause, T., Faust, D. and Fuchs, M.: Technical Note: *n*-Alkane lipid biomarkers in loess: post-sedimentary or syn-sedimentary?, Discussions, Biogeosciences, 9, 9875–9896, doi:10.5194/bgd-9-9875-2012, 2012.

---

## Author Response (AR1)

Dear Marcel van der Meer, thank you for serving as handling author of our manuscript and for the constructive feedback. Please find attached the revised versions of the author responses to referee #1 and #2 as well as the manuscript and the supplements (track change marked versions).

Best regards,
Johannes Hepp

**Revised reply to Referee #1**

by Johannes Hepp, Michael and Roland Zech & co-authors

We are grateful to anonymous Referee #1 for her/his constructive suggestions helping to improve our manuscript. Please find our replies to the individual comments below.

*Major issues:*

*1) The brGDGT calibration presented here is of limited use, since the study uses an outdated method to measure brGDGTs and does not distinguish between the 5 methyl and 6 methyl compounds. Hepp et al. thus calibrate indices (CBT and MBT') that have fallen out of favor and been replaced by the more robust CBT' and MBT5Me indices. The new indices and new methods developed by De Jonge et al. (GCA, 2014, doi: 10.1016/j.gca.2014.06.013) and Hopmans et al. (Organic Geochemistry, 2016, doi: 10.1016/j.orggeochem.2015.12.006) are not even mentioned in the text, and the limitations of the brGDGT data presented here are not acknowledged. Without reanalyzing these samples with a method that resolves all isomers, I fear that the present sample set has limited value for the calibration of brGDGT-based proxies.*

→ Referee #1 is right in his/her statement that the GDGT data presented in our manuscript were not acquired based on the up-to-date method. During revision, we therefore explicitly emphasize that meanwhile new indices and methods were developed (including citations). We would still see a high value of having our GDGT dataset published. De Jonge et al. (2014) presented a new HPLC method which enables the separation for the brGDGTs with m/z 1036, 1034 and 1032, 1050, 1048 and 1046 into 6-methyl and 5-methyl stereoisomers. The old method did not allow such a separation (Zech et al., 2012b) thus in the calibration often the sum of 6 and 5-methlyted brGDGTs was used because the shoulders of the peaks could not be identified in each case (see and compare De Jonge et al., 2014; Peterse et al., 2012). This introduce scatter to the MBT´-CBT-based MAT reconstructions and can cause a correlation between pH and MBT` (for more details see De Jonge et al., 2014). The authors moreover show that the 6-methyl brGDGTs are ubiquitous abundant in soils from all over the world. However, they also compare reconstructed MAT values based MBT´-CBT calibration (Peterse et al., 2012) and their new developed MAT$_{mr}$ calibration and state that they plot around a 1:1 relationship. They furthermore state that only for arid areas a strong deviation can be obtained. Finally, they conclude that the use of the new developed calibrations will improve the MAT and pH reconstructions for dry conditions/areas. Because our study transect spans form southern Germany to southern Sweden, representing temperate and humid climate conditions, we argue that the usage of the older HPLC method do not introduce a systematic error in our reconstructions. Still, a higher variability/scatter is associated with the calibration of Peterse et al. (2012) and therefore present in our MAT and pH reconstructions. However, we firstly compared our data only to those of Peterse et al. (2012), and we secondly prevented an over-interpretation of our data. This discussion is now included as a separate discussion chapter in the revised version of the manuscript.

*2) There are some big assumptions in the proposed approach for reconstructing relative humidity using paired δ2H values of n-alkanes and δ18O values of sugars. In particular, the assumption that biosynthetic fractionation for these compounds is constant is contradicted by lots of existing work, which is briefly mentioned by the authors in their discussion. Figure 8 is not a very good advertisement for the utility of the paired δ2H-alkane/δ18O sugar approach, and the lack of correlation suggests that some of the many assumptions that go into this method are not valid. This paired approach has not caught on beyond the Zech group, and the data presented here suggests that it may not be useful as presently conceived. The authors state they have shown the "great potential" for this proxy. I remain unconvinced by the data and analysis shown here.*

→ We accept that Referee #1 remains unconvinced by our coupled $\delta^2H_{n\text{-alkane}}$-$\delta^{18}O_{sugar}$ biomarker approach. We moreover we (i) agree, (ii) are aware and (iii) explicitly state that the assumption of constant biosynthetic fraction is likely a major uncertainty of our approach. Still we are convinced that the 'opening of the second dimension' by our group is a cutting-edge step forward and more promising than focusing on $\delta^2H_{n\text{-alkane}}$ alone. The reason for other working groups not having caught on the coupled approach might have to be seen, in our opinion, in the uniqueness of compound-specific $\delta^{18}O$ analyses: according to our knowledge, only 3 working groups world-wide have respective experience/publication records. Still, we would be delighted to see the coupled approach being tested or applied by other groups, readily in cooperation with us. Still we acknowledge that focusing on $\delta^2H_{n\text{-alkane}}$ hast the advantage that a lot of research was done and many working groups around the world published results during the last years. The coupling is still work in progress but we think we have to start somewhere and this introduces also (new) uncertainties for sure, but is still worth to publish and start the process of proxy improvement via scientific discussions with this.

Possibly, Referee #1 misunderstood Fig. 8. No correlation for the data points shown in Fig. 8 are to be expected. We clarified in our revision that Fig. 8 illustrates the 'concept of the coupled $\delta^2H_{n\text{-alkane}}$-$\delta^{18}O_{sugar}$ biomarker approach'. This conceptual figure illustrates (together with Fig. 9) that $\delta^2H/\delta^{18}O_{prec}$ values reconstructed by the coupled approach are more accurate than $\delta^2H_{prec}$ values reconstructed using $\delta^2H_{n\text{-alkane}}$ alone. Moreover, Fig. 10 illustrates that reconstructed RH values under deciduous forest sites and grassland sites are quite well in accordance with RH values of climate stations, thus indeed demonstrating the great potential of the coupled approach.

*3) The writing is in places unclear and difficult to follow. I have noted a few of these instances in my technical corrections, but the manuscript would benefit from more careful editing.*

→ We insure a technical and grammatical improvement for the revised version of the manuscript.

*Specific comments:*

*Line 110: This adds up to more than 16, some sites were considered to be more than one of these categories? Would be good to rewrite to clarify*

→ Following the recommendation of Referee #1 we will restructure this sentence. The revised version will read: "In November 2012, we collected 29 topsoil samples (0-5 cm depth) from 16 sites along a transect from Southern Germany to Southern Sweden (Fig. 1A). We distinguished between coniferous forest (con, n = 9), …".

*Line 114: Was there a threshold for what was considered "close-by"?*

→ We agree with Referee #1 that this was not obvious so far in the manuscript and especially not in the supplementary material where the longitude, latitude and altitude were provided for the climate stations (Tab. S2) but not for the locations/sites. In the revised manuscript, we will add the respective characteristics to Tab S1.

*Line 133: Machine learning techniques like random forest aren't so commonly used in Biogeosciences and it would be helpful to provide more details here. How many trees did you use? How was data partitioned into training and testing sets? What metric was used to assess model performance? What was the minimum number of samples in the terminal nodes? What was the maximum number of terminal nodes? What variables ended up being ranked as most important (could be useful to show a plot of ranked variable importance in the supplemental materials)?*

→ As suggested, we will add a supplementary method description part and refer to it in the text.

*Line 136: Why wasn't it possible? Lack of measured data for a robust training data set? Please specify*

→ Because no precipitation isotope data was available for the Danish and Swedish sites.

*Lines 128-139: How did the calculated values you obtained for the German sites compare to OIPC? What is your evidence for your approach providing superior estimates of precip isotopes than OIPC? OIPC is obviously not perfect, but as written, we have no evidence to evaluate if your results are any more accurate. There is also no discussion of the implications of using one target for precip isotopes in the southern half of your transect and a different one in the northern half.*

→ Please allow us to refer to the (cited) Diploma Thesis of Schlotter (2007): there are numerous reasons mentioned already in the introduction highlighting that OPIC is probably not the most robust estimator for middle and high latitudes. That's why we used our own regionalization where it was possible.

*Section 2.3.1: No internal standard was added? How do you account for losses of brGDGTs during sample handling?*

→ We used standard laboratory procedure for GDGT sample preparation. The internal standard was, as written, added before the measurements. A correction for GDGT losses during sample preparation is therefore not possible.

*Lines 165-171: This is not the most current method used for robust brGDGT analysis (see Hopmans et al., Organic Geochemistry, 2016. DOI: 10.1016/j.orggeochem.2015.12.006). Does your method allow for 5'*

*and 6' methyl brGDGTs to be distinguished from one another? If not, severely diminishes the accuracy of results. Based on the results that are shown, it seems like this method does not distinguish the different isomers.*

→ That's correct. Please see our reply to major issue 1.

*Lines 172-173: how was the pH measured?*

→ We will include the information that a pH meter was used.

*Section 2.3.2: Were the n-alkanes quantified prior to measuring their stable isotopes?*

→ Yes, namely by Schäfer et al. (2016). We therefore added the following sentence in the section: "For more details about *n*-alkane quantification the reader is referred to Schäfer et al. (2016). ".

*Also, please briefly describe the operating conditions of the GC-pyr-IRMS (or cite another publication that used an identical method and provides all the relevant details)*

→ As suggested, we added now in the revised version of the manuscript a reference (Christoph et al., 2019), in which the method is described in more detail and we added that the $^2$H pyrolysis reactor temperature was kept at 1420 °C.

*Line 199: It is not clear how you had 29 samples from 16 sites. Were some of the sites sampled in duplicate?*

→ We will clarify during revision that 29 samples were collected from 16 sites. These are, however, no duplicates, but rather different dominant vegetation types (see reply above).

*Lines 211-221: The more robust indicator of soil pH is CBT' and the more robust indicator of soil temperature is MBT5me (De Jonge et al., GCA, 2014, DOI:10.1016/j.gca.2014.06.013).*

→ See our reply to major issue 1.

*Lines 227-229: A number of papers have shown that ebio is not constant and different among plant types and seasonally. See for example Feakins & Sessions 2010 (cited previously), Eley et al., GCA, 2014 (DOI: 10.1016/j.gca.2013.11.045), Cormier et al., New Phytologist, 2018 (DOI: 10.1111/nph.15016).*

→ That's true and especially important when only $\delta^2H_{n-alkane}$ is used to reconstruct $\delta^2H_{leaf-water}$. Nevertheless, we emphasize in our manuscript that $\varepsilon_{bio}$ is a major uncertainty in our coupled approach, too. At the same time, it's exactly such uncertainties why we need climate transect calibration studies as the one presented here for Europe.

*Lines 383-385: are these concentrated weighted means? That is what is typically used to compare d2H values of n-alkanes where not all homologues are present in all samples*

→ We used here mean values, because the areas and concentrations where not determined during isotope measurements.

*Line 395: I think you mean "unenriched xylem water"?*

→ Yes, changed.

*Lines 431-432: This is not particularly convincing, the reconstructed precipitation isotopes are not correlated with the GIPR/OIPC precipitation isotopes. No evidence is provided to show that this approach is any better than the most up to date methods for obtaining precipitation isotopes from leaf wax n-alkane isotopes alone. For example, how do your results compare to the predictions from the proxy system model developed by Konecky et al. (JGR-Biogeoscience, 2019, DOI: 10.1029/2018JG004708)? Maybe your approach is better, but you need to prove this by providing a direct comparison, rather than just telling us*

→ Please note that we do not necessarily expect a good correlation of our reconstructed $\delta^2H/\delta^{18}O_{prec}$ values with the GIPR/OIPC data, but rather a good (accurate) match on the 1:1 line. Nevertheless, many thanks for pointing us to the new publication by Konecky et al. (2019). While we will readily include a respective citation, we think that a direct comparison of our approach with the one suggested by Konecky et al. (2019) would be beyond the scope of our manuscript.

*Lines 448-450: If this was the case, wouldn't you expect all the coniferous sites to be biased in the same direction? Instead, they are evenly distributed above and below the 1:1 line*

→ No, please see Fig. 9: we do not see that the coniferous sites are evenly distributed around the 1:1 line. Except for one data point, they are clearly below the 1:1 line.

*Line 454: Is this signal damping correction shown anywhere? How would this work practically in sediments?*

→ No, sorry. This signal damping correction is not shown or quantified in this manuscript. This would require a quantitative estimation of the contribution of grass vegetation to the total biomass pool in the topsoil. For an example how such a correction can be applied to lake sediments please see e.g. Hepp et al. (2019, CP).

*Lines 467-468: Actually, there are plenty of n-alkanes in roots and they have very different H isotopic composition than in leaves. See work from Guido Wiesenberg's group and Gamarra and Kahmen. I'm also confused about what you are referring to as "the discussion". There is not a separate discussion section to this manuscript.*

→ Changed to "Zech et al. ,2012b and the discussion therein". We do not agree and we are not aware of any new studies showing that *n*-alkanes are produced in large amounts by roots in comparison to leaves. Recent studies show (e.g. Gamarra and Kahmen, 2015) that root *n*-alkane concentration is always the lowest compared to the other plant tissues sampled.

*Lines 489-494: Not stated here is that there is no correlation between the reconstructed and measured RH values. This suggests that this approach for reconstructing RH is not particularly useful Line 565: The data in the paper is not very convincing that there is great potential for the coupled d2H n-alkane d18O sugar approach*

→ We think this is connected to the low range of measured RH values along this European climate transect and the uncertainties of the coupled approach for reconstructing RH values.

Therefore, the lack of a respective correlation is explainable. Please compare a similar climate transect study by Tuthorn et al. (2015, BG) where the RH range is much larger and where indeed a significant correlation can be found. For this European transect study here, the usefulness of the coupled approach for reconstructing RH values should be rather inferred from the quite well 1:1 match for deciduous forest sites and grassland sites (cf. Fig. 9). The RH underestimation for coniferous forest sites can be easily explained with the extremely low *n*-alkane production of coniferous trees (see ll. 495-502).

*Lines 566-567: I don't see evidence of this in your analysis, nor examples of how you would take vegetation into account when applying this proxy.*

→ See for example Hepp et al. (2019).

*Technical corrections and typing errors:*

*Lines 54-56: The way this sentence is written is confusing. Suggest rewriting as "Climate proxies based on molecular fossils, also known as biomarkers, have great potential...*

→ Changed.

*Line 56: don't need the comma after "particular"*

→ Changed.

*Line 59: "need to be known"?*

→ Changed.

*Line 61: It would be better to start this paragraph with a clear link back to the previous one*

→ We now start the paragraph with "One famous and widely applied lipid biomarker group are terrestrial branched glycerol dialkyl glycerol tetraethers (brGDGTs). They are synthesized… and…"

*Line 74: don't need commas before and after "it is known"*

→ Changed.

*Line 79: Again, some sort of transition would be helpful to begin this paragraph*

→ We now start the paragraph with "Concerning paleohydrology proxies, compound-specific…"

*Line 82: "all along the way" too wordy*

→ Changed

*Lines 93-94: "as well as concerning possible effects related to" awkward phrasing*

→ Changed.

*Figure 1: would be nice to have a legend on panel B or have the axis colors match the variable colors. At the moment we are left to guess that blue bars are precip and the red dots are temp, since this is not stated in the figure caption or the legend. Also would be nice to offset the panel letters with a () or . to break them apart from the title of the panel*

→ Changed.

*Line 180: No "the" needed in front of ETH*

→ Changed.

*Line 225: the n at the beginning of n-alkane should be italicized. Check throughout*

→ Changed.

*Line 234: Generally, figures should be numbered in the same order that they are referenced in the text*

→ Checked and changed if necessary.

**Revised reply to Referee #2**

by Johannes Hepp, Michael and Roland Zech & co-authors

*GENERAL:*

*The topic of the manuscript is interesting and important as it deals with the evaluation of highly promising proxies used to reconstruct past environmental conditions. While the data produced are rare and are certainly worth publishing, the manuscript has severe flaws that prevent, in my opinion, its publication in this form.*

→ While we are grateful to Referee #2 for her/his constructive suggestions helping to improve our manuscript (see our replies below).

*MAJOR PROBLEMS:*

*A) While reading the manuscript, the connection between GDGT and the plant proxies (i.e. n-alkanes and hemicellulose) is not clear and seems disconnected as if from two separate manuscripts. Moreover, in the section 3.1 of the discussion, the GDGT data are presented in a way leading the readers to believe that these molecules are produced by plants.*

→ Thank you for raising this issue. We think that approaches are based on biomarkers/molecular proxies and are used for paleoclimate reconstructions. Moreover, we clearly state and explain in the introduction and method sections how the applied biomarkers (GDGT´s as well as *n*-alkanes and sugars) are produced, how calculations are done and how the proxies can be interpreted. Please note that there are plenty of studies in the literature presenting both GDGT and $\delta^2H_{n\text{-alkane}}$ results in one publication. However, we will check the whole manuscript during revision in order to be clear about the origin of the presented biomarker proxies.

*B) The other major point is that the authors suggest that it is "often" not feasible to disentangle between the evapotranspirative enrichment from the precipitation signal, but there is at least another well-established method to do so and published in Climate of the Past (see recent Sachse's group publications, e.g. A dual-biomarker approach for quantification of changes in relative humidity from sedimentary lipid D/H ratios, Climate of the Past, 2017). While this method should at least be mentioned, I also believe the method should be compared to help the readers understand the full set of tools available to study that issue. These two methods are very likely to be highly complementary.*

→ Thank you for raising this issue, but please note that the 'dual biomarker approach' of Rach et al. (2017, CP) is not applicable to terrestrial (soil) samples/archives, it works only under lacustrine settings. For a critical evaluation and assessment of both approaches when applied to lacustrine paleoclimate archives, we kindly refer our readers to Hepp et al. (2019, CP) and to our replies to the referee and short comments ([https://www.clim-past.net/15/713/2019/cp-15-713-2019-discussion.html](https://www.clim-past.net/15/713/2019/cp-15-713-2019-discussion.html)).

*SPECIFICS:*

*Line 298 to 303: This section is not clear due to some typos or mistakes, please reformulate.*

→ Changed.

*Line 389 to 407: While the difference of ebio is reported at the end of the section (around line 477 to 487), the possibility that a variable ebio could explain the different signals in different types of vegetation, beside the damping effect, is evacuated of the discussion. This should at least be discussed.*

→ Changed.

*Line 432: Is that referring to simply using isotope values of a single compound? What is that hitherto method (reference missing?)? I believe this brings us back to the problem B. The results would gain a lot to be compared with the updated tool box of proxies.*

→ The sentence was slightly changed. See also our reply to 'major problem B'.

*Line 444 to 458: The argumentation is not clear/convincing, please reformulate.*

→ We deleted the respective sentence from the revised version of the manuscript.

*Line 483-484: The idea of a variable ebio is well expressed in general, but references to some recent works is missing that shows even greater variability in n-alkane dD values under different metabolisms (e.g. Cormier et al, 2018 – New Phytologist, Tipple & Ehleringer 2018 – Oecologia, Cormier et al, 2019 – Oecologia)*

→ Please note that we already included Cormier et al. (2018) in the actual version of the manuscript and that the fact is mentioned that $\varepsilon_{bio}$ can range even larger when also the metabolic status of the plants is considered. However, we changed the respective sentence to: "The wide range in biosynthetic $^2H$ fractionation factors, which can be even larger, is therefore also related to the carbon and energy metabolism state of plants (Cormier et al., 2018).".

*Line 490 to 494: Please reformulate, this section is not clear.*

→ We changed the quoting of Fig. 10B.

*Line 550: If the author are really considering a variable ebio, the damping effect can only potentially explain the different signals observed in different types of vegetation. Again, ebio should be part of the points because standing alone, they can induce confusion even if mentioned afterward.*

→ You are right. Gao et al. (2014) and Liu et al. (2016) showed that the $\varepsilon_{bio}$ of monocot plants could larger than those of dicots. This would therefore course a more negative apparent fractionation factor for grasses compared to trees. We observe that the apparent fractionation is indeed more negative for the grass sites compared to the forest sites. We will included a discussion about the indistinguishable effects of "signal damping" vs. variable $\varepsilon_{bio}$ along with vegetation types in the respective parts of the manuscript.

[revised manuscript text omitted]

n.a. = not available

---

## Author Response (AR2)

**Reply to the Comments to the Author**

by Johannes Hepp & co-authors

Dear Marcel van der Meer, first of all, I want to express my gratefulness about your constructive handling of our manuscript and that we had the possibility to revise also our replies the referees. Second, I want to thank you for raising the constructive suggestions in the comments to the authors. Please find our replies to your issues below.

*First of all I would like to thank you for this reply to the reviewers comments, much better than the original reply. I also want to thank the reviewers again for their reviews. I totally agree with you that the limited amount people doing this combined approach should not affect your development and application of the proxy in anyway. That being said, I do see the point of the reviewers as well. I guess the measurements are quite tricky and you have to make quite a few assumptions, like a fixed εbio, to make this work. All and all maybe not so convincing yet for other users to get involved in, with the emphasis on yet. I do see the advantage of this combined approach. Of course hydrogen and oxygen behave very similarly with respect to humidity, temperature, evaporation etc. and I can't really judge if that is good or bad. There are benefits to independent proxies, say hydrogen isotopes from organic molecules and oxygen isotopes from inorganic carbonates for instance.*

*Of course you did measure completely independent proxies for temperature and pH, but these are not linked to the isotope results in anyway? Is there anything you say based on the isotope data that can be back-up by the GDGT data? Something about the local (micro) climate being different from that of the weather stations, for instance? The GDGTs were measured on the same samples as the isotopes, they should reflect the same "micro" climate. Or the underestimation of the RH, can that be linked/supported by some of the other results? Just to link the two independent methods a bit more together?*

➔ Thank you for raising that important issue and for you suggestions how to link the both proxy types presented in the study closer together. Since the GDGTs are produced (or originate) mainly from microorganisms in the soil and the (long-chain) *n*-alkanes and the (hemicellulose) sugars are known to be produced by plants (mainly higher terrestrial plants when considering the homologues/compounds which we focus on), it is not that simple to find connections between those datasets, besides they are synthesized in the same environment. That's the reason why we decide to interpret the proxy results separately in the current manuscript. However, when looking on the correlation between reconstructed and "measured" $T_{MA}$, the relationship based on the data achieved from our study would be rather weak. But when the our data is plotted together with the large (world wide) dataset from Peterse et al. (2012), our data fits well to the overall correlation (Fig. 6A). The reconstructions based on the isotope data have therefore to tackle the same issue regarding a rather small data set and rather medium changes along our transect with regard to the climate conditions. That's probably a link between the GDGT-based reconstructions and the isotope-based once. We will add a respective sentence to the conclusion to have a connection between the GDGT part and the isotope part. However, besides the correlation between reconstructed source water isotope composition and RH to measured values is rather weak due to the rather small range which covers the transect data, the main issue here is that the data plot partly not well on the 1:1 line, which should be the case if the idea behind the proxy calculations are right. That's why we intensively discuss the influence of different vegetation types on our isotope proxy results. That's still something what does obviously not affect the GDGTs, which is also interesting. We will add therefore a respective sentence to highlight this finding in the conclusion as well as in the abstract.

*The other thing I was thinking about is that, since the water isotopes behave so similarly you could use the combined measurements to examine , for instance, εbio. Flip your proxy around so to say. Plenty of options for future work, I think.*

➔ Thank you for raising that inspiring issue. We think that would be a good project for a further study because it would be nice to combine here all available compound-specific isotope (terrestrial *n*-alkanes as well as sugars) and try to parameterize the model based on measured data. One could for example use a kind of model presented by Konecky et al. (2019) and expand it to (hemicellulose) sugars and cellulose.

*I have a few minor comments; You mention in your response and in the discussion that you cannot see the difference between damping or differences in εbio, this has been left out of the abstract?*

➔ Thank you for that constructive comment. That was just a mistake. We will add this as it is mentioned in the conclusions.

*In the introduction you mention quite a few variables on which the hydrogen isotopic composition of n-alkanes depend, some of these are what you would like to reconstruct, so that is not a bad thing right?*

➔ Yes, you are right. All those factors are included in the isotope equations and thus in the isotope proxy calculations/reconstructions.

*In the figure 1 legend you mention that the map is sourced by the US National Park Service (just checking)?*

➔ Yes, that's correct.

In line 351 after the, it says "precipitation drops …. " I don't see how this is linked to the rest of this sentence?

➔ You are right, this is now corrected to "when precipitation amount drops below 700-800 mm."".

**Literature**

[revised manuscript text omitted]